# Genus-wide sequencing supports a two-locus model for sex-determination in *Phoenix*

Maria F. Torres[1,2], Lisa S. Mathew[3], Ikhlak Ahmed[1], Iman K. Al-Azwani[3], Robert Krueger[5], Diego Rivera-Nuñez[6], Yasmin A. Mohamoud[3], Andrew G. Clark[7], Karsten Suhre [4] & Joel A. Malek [1,3]

The date palm tree is a commercially important member of the genus *Phoenix* whose 14 species are dioecious with separate male and female individuals. To identify sex deter-mining genes we sequenced the genomes of 15 female and 13 male *Phoenix* trees representing all 14 species. We identified male-specific sequences and extended them using phased single-molecule sequencing or BAC clones. We observed that only four genes contained sequences conserved in all analyzed *Phoenix* males. Most of these sequences showed simi-larity to a single genomic locus in the closely related monoecious oil palm. CYP703 and GPAT3, two single copy genes present in males and critical for male flower development in other monocots, were absent in females. A LOG-like gene appears translocated into the Y-linked region and is suggested to play a role in suppressing female flowers. Our data are consistent with a two-mutation model for the evolution of dioecy in *Phoenix*.

[1] Department of Genetic Medicine, Weill Cornell Medicine-Qatar, PO Box 24144, Doha, Qatar. [2] Department of Biological Sciences, University of Cincinnati, Cincinnati, 45221, OH, USA. [3] Genomics Laboratory, Weill Cornell Medicine-Qatar, PO Box 24144, Doha, Qatar. [4] Department of Physiology and Biophysics, Weill Cornell Medicine-Qatar, PO Box 24144, Doha, Qatar. [5] USDA-ARS National Clonal Germplasm Repository for Citrus & Dates, Riverside, 92507, CA, USA. [6] Department of Plant Biology, Faculty of Biology, University of Murcia, Murcia, 30100, Spain. [7] Department of Molecular Biology and Genetics, Cornell University, Ithaca, 14853, NY, USA. Correspondence and requests for materials should be addressed to J.A.M. (email: jom2042@qatar-med.cornell.edu)

The origin and evolution of separate sexes is a subject that has long intrigued biologists. The presence of separate male and female individuals (dioecy) ensures certain advantages such as outcrossing and improved fertility by allowing allocation of resources to evolve independently in the two sexes[1]. One model for the transition from cosexuality to dioecy involves two successive mutations in a pair of chromosomes: a recessive male-sterility mutation that creates females (initiating a proto-X chromosome) and a dominant female sterility mutation in a gene on the proto-Y chromosome creating males. Recombination may subsequently become suppressed between these two genes[2,3]. Lack of recombination in the heterogametic sex (XY) permits the accumulation of mutations and repetitive elements, which in turn leads to chromosome degeneration (reviewed in ref. [4]). While most species of animals have separate sexes, only 5–6% of all plant species are dioecious[5]. Various genomic approaches[6] have allowed the identification of sex-linked genes in papaya (*Carica papaya*), white campion (*Silene latifolia*), grapevine (*Vitis vinifera*), poplars (*Populus* sp.), and the herb *Mercurialis annua* and candidate sex determining genes have been identified in asparagus (*Asparagus officinalis*) and persimmon (*Diospyros virginiana*)[7–12]. Information about how these genes control plant sex remains elusive although two mutations appear to have been involved in both asparagus and persimmon[12–14].

The female date palm (*Phoenix dactylifera*) is another important dioecious crop, producing edible dates and having high agricultural value in North Africa, the Middle East, and South East Asia. We previously genetically mapped a sex-linked region to the long arm of linkage group 12 and estimated this region to span 13 Mb, or 2% of the genome[15,16] but leaving sex determination genes to be identified[15,17,18]. Phylogenetic analysis of one sex-linked gene suggested that dioecy evolved before speciation in the genus *Phoenix*[18] consistent with the fact that all 14 members of the genus *Phoenix* are dioecious, unlike related genera in the palm family[19]. Furthermore, hybridization can occur among species with fertile progeny in some cases. Hypothesizing a single origin of dioecy, we designed a study that combines de novo whole-genome sequencing and comparative genomics across all 14 members of *Phoenix* with the aim of identifying the genes responsible for sex determination in the genus. The study is based on the hypothesis that genes for male function and female organ suppression in the ancestral *Phoenix* would be maintained on the Y chromosome in present-day species. We identify four candidate genes, of which three are completely absent in all female *Phoenix* individuals, but present in two closely related hermaphroditic palms. We discuss the putative role of these genes and propose a model for the origination of dioecy in *Phoenix*.

## Results

**Identification of male-specific sequences**. Identification of 16 bp kmer sequences unique to either males or females in each species (Table 1 and Supplementary Table 1) revealed male-specific kmers in all 13 species tested (no *P. pusilla* male could be identified for sequencing) (Fig. 1a). In total, we found 1653 kmers (Supplementary Data 1) present in all 13 males and absent from all 14 females tested. In contrast, no kmers were unique to all females; kmers present in females but absent in males were not shared by more than eight species of the genus, consistent with an XY sex determination system (male heterogamety) (Fig. 1a).

**Full-length sequencing of male-specific sequences**. Sequencing reads derived from a male date palm and that carried the male-specific kmers were assembled and used to design probes to identify and then sequence clones from a male date palm BAC library. Where possible, BACs identified as having male-specific

### Table 1 Genomes sequenced in this study

| Species[a] | Sex | Collection | Short name | Sequence coverage (fold) |
|---|---|---|---|---|
| *P. dactylifera* (Deglet Noor) | F | USDA, CA | dnPdF | 71.4 |
| *P. dactylifera* (Deglet Noor BC5) | M | USDA, CA | dnPdM | 67.5 |
| *P. dactylifera* (Khalas) | F | Qatar | khlsF2016 | 69.5 |
| *P. roebelenii* | F | USDA, CA | P03roeF | 23.3 |
| *P. roebelenii* | M | USDA, CA | P5roeM | 37.2 |
| *P. canariensis* | F | USDA, CA | P08canF | 34.8 |
| *P. canariensis* | M | USDA, CA | P09canM | 67.1 |
| *P. hanceana* (loureiroi) | M | USDA, CA | P13hanM | 28.1 |
| *P. hanceana* (loureiroi) | F | USDA, CA | P14hanF | 27.5 |
| *P. paludosa* | F | USDA, CA | P17palF | 38.5 |
| *P. paludosa* | M | USDA, CA | P19palM | 45.7 |
| *P. reclinata* | M | USDA, CA | P20recM | 58.1 |
| *P. reclinata* | F | USDA, CA | P21recF | 41.4 |
| *P. sylvestris* | F | USDA, CA | P23sylF | 32.4 |
| *P. sylvestris* | M | USDA, CA | P25sylM | 27.6 |
| *P. andamanensis* | F | UMH, Spain | Q01andF | 36.7 |
| *P. andamanensis* | M | UMH, Spain | Q03andM | 41.8 |
| *P. caespitosa* | F | UMH, Spain | Q07caeF | 55.0 |
| *P. caespitosa* | M | UMH, Spain | Q09caeM | 45.8 |
| *P. acaulis* | F | UMH, Spain | P02acaF | 31.0 |
| *P. acaulis* | M | UMH, Spain | Q10acaM | 29.1 |
| *P. theophrasti* | F | UMH, Spain | Q17theF | 33.8 |
| *P. theophrasti* | M | UMH, Spain | Q19theM | 93.3 |
| *P. atlantica* | F | UMH, Spain | Q22atlF | 48.1 |
| *P. atlantica* | M | UMH, Spain | Q23atlM | 49.0 |
| *P. rupicola* | F | UMH, Spain | P0XrupF | 23.0 |
| *P. rupicola* | M | UMH, Spain | Q15rupM | 36.4 |
| *P. pusilla* | F | UMH, Spain | Q14pusF | 31.0 |
| *Brahea dulcis* | Herm | Huntington Gardens, CA | braheaSpp | 45.0 |
| *Livistona rotundifolia* | Herm | Huntington Gardens, CA | livistonaSpp | 32.5 |

[a]*P.* denotes *Phoenix*

sequence were extended by further probing with sequences from the end of the BACs until repetitive sequence prevented further extension. Additionally, phased single-molecule sequencing scaffolds (10X Genomics) of the same species' male genome were searched for male-specific kmers. The phased 10X Genomics assembly included two haplotypes of 25,342 scaffolds spanning 514 Mb with an N50 scaffold length of 324 kb comparing well with previous un-phased assemblies of the date palm genome[20].

All 1653 male-specific kmers were within just three BAC contigs (dpBGPATlike, dpBCYPlike, dpBLOGlike) and one phased sequence scaffold (dpB3Y) summing to a total of 913 kb (Fig. 1b).

To understand the region(s) surrounding these genus-wide, male-specific sequences, and to identify X-linked counterparts to any Y-linked sequences, we further probed the same BAC library and 10X Genomics scaffolds with sequences carried by the males of at least three species. This yielded 11 BAC contigs and 30 phased sequence scaffolds spanning ~5.5 Mb (~11 Mb of male and female alleles) in the region of interest (Supplementary Table 2). BAC contigs and sequence scaffolds that contained kmers conserved in some but not all males of the genus likely represent the evolution of further regions of recombination

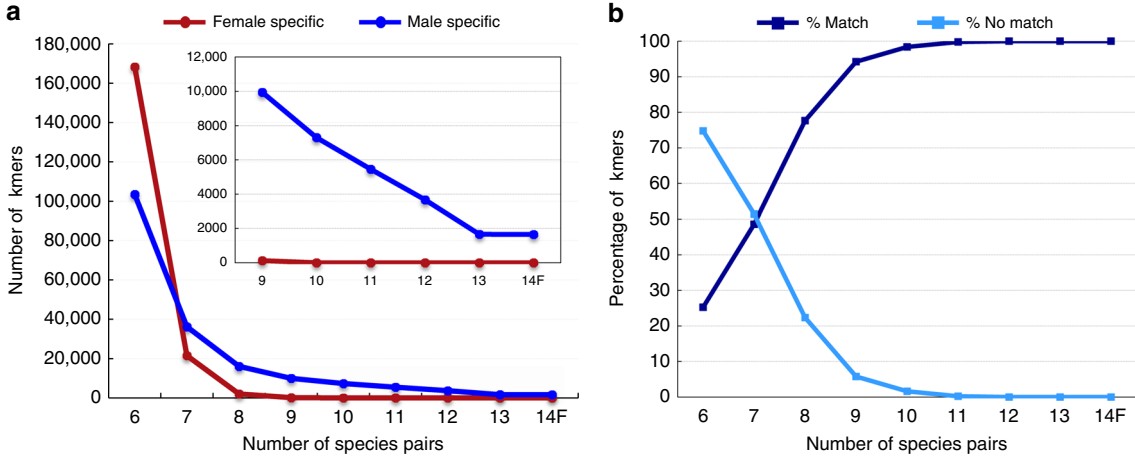

**Fig. 1** Identification of male-specific kmers in the genus *Phoenix*. **a** Sixteen base pairs kmers identified only in the male (blue) or female (red) individual for each species were tested for their presence in other species. As expected for an XY sex determination system, kmers specific to males maintained counts through all species tested while female-specific kmers disappeared after requiring presence in more than eight species. **b** Kmers specific to male *Phoenix* enumerated against the male-specific BAC contigs. Kmers specific to males in at least 12 species are fully covered by the sequenced BACs in this study (dark blue—% kmers matching BACs, light blue—% kmers not matched to BACs). Only a female was available for *Phoenix pusilla* comparisons (14F)

suppression between the Y and X chromosomes in some species (new evolutionary strata).

**Annotation of male-specific sequences**. Annotation of all four contigs with genus-wide male-specific kmers (Fig. 2) revealed just a few genes surrounded by highly repetitive sequence. These were full-length genes with similarity to *CYP703*, *GPAT3*, *LOG* and, to a lesser extent, cytidine deaminase (for which we found a male and female allele). The BAC contig containing cytidine deaminase also included a degraded copy of the MAP1-like gene that included no male-specific kmers.

Searches of the NCBI database indicated that one of the male-specific genes is a Cytochrome P450 (CYP) with 91% DNA sequence identity to the oil palm (*Elaeis guineensis*) *CYP703A* gene (LOC105059962) (Supplementary Fig. 1). While angiosperm CYP families often have many paralogs, the CYP703 family is a single member gene family[21]. A phylogenetic analysis of the predicted CYP-like proteins from *Phoenix*, and the two hermaphroditic palms *Brahea* and *Livistona* plus CYP450 proteins from 11 other plant species[21,22] indicated that the CYP gene identified here belongs to the CYP703 family and is a putative ortholog of *CYP703A3* from rice (not shown). We found no evidence of complete or partial sequences of *CYP703* in the published female reference genomes or the transcriptome data available from date palm[15,20].

The second male-specific gene identified also has 91% DNA sequence identity to an oil palm glycerol-3-phosphate acyltransferase 6-like (GPAT6-like) gene (LOC105059961), and phylogenetic analysis suggests that it belongs to the GPAT 1/2/3 (annotated GPAT3) clade[23] (Supplementary Fig. 2).

Normalization of the sequence read coverage to genome-wide coverage indicated that both *CYP703* and *GPAT3* are haploid (1N) in males (Fig. 3d) supporting the hypothesis that these are indeed unique to the Y chromosome. The third male-specific sequence is similar to a family of genes termed *Lonely-Guy* (LOG). Our phylogenetic analyses suggested that the LOG-like gene is a paralog of the date palm autosomal gene Pd_LOC103701078 that was also covered by BAC sequencing in this study (Supplementary Figs. 3 and 4). Based on synteny, this autosomal gene appears to be the ortholog of the oil palm chromosome 12 gene, LOC105055182, which is similar to the rice genes *OsLOG5* and *OsLOG9*. The finding of two copies in the date

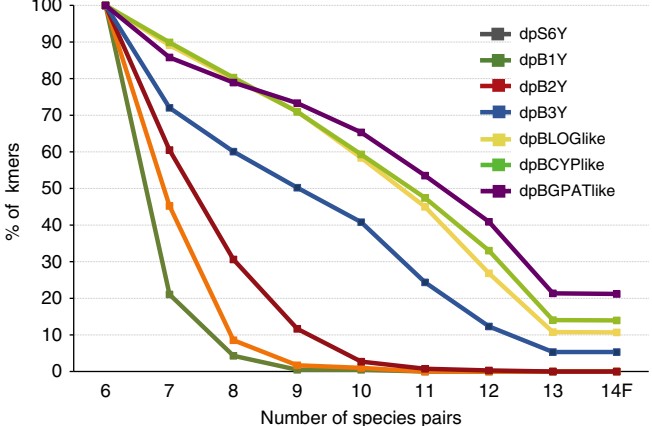

**Fig. 2** Male-specific BAC and scaffold sequences contain varying numbers of male-specific kmers. Percent of kmers matching in six species were used to normalize to 100% for size of the sequence contig. The dpBGPAT3like contig (purple) maintained the largest percentage of kmers through the 14 species, followed by dpBCYPlike (light green), dpBLOGlike (yellow), and the dpB3Y scaffold (blue). For comparison, three scaffolds (green, orange, red) containing kmers conserved in only a few males of the genus were included (dpS6Y, dpB1Y, dpB2Y). These are likely due to the spread of non-recombination on the Y chromosome after speciation as they do not maintain kmers in all 14 species. For comparison, see genes synonymous mutation rates in the text. Only a female was available for *Phoenix pusilla* comparisons (14F)

palm suggests that the date palm male-specific (i.e. sex-linked) copy arose by a duplication of the autosomal gene after divergence of *Elaeis* and *Phoenix*. We identified possible orthologs of the male-specific LOG kmers in the hermaphroditic palms *Brahea* and *Livistona*, suggesting that this duplication occurred in a common ancestor of the subfamily Coryphoideae after the split from *Elaeis*. The exon sequences of the autosomal and Y-linked paralogs are highly similar (Supplementary Fig. 4), but large numbers of male-specific kmers were found in introns and putative regulatory regions of the gene, making the two loci easily distinguishable (Fig. 3d).

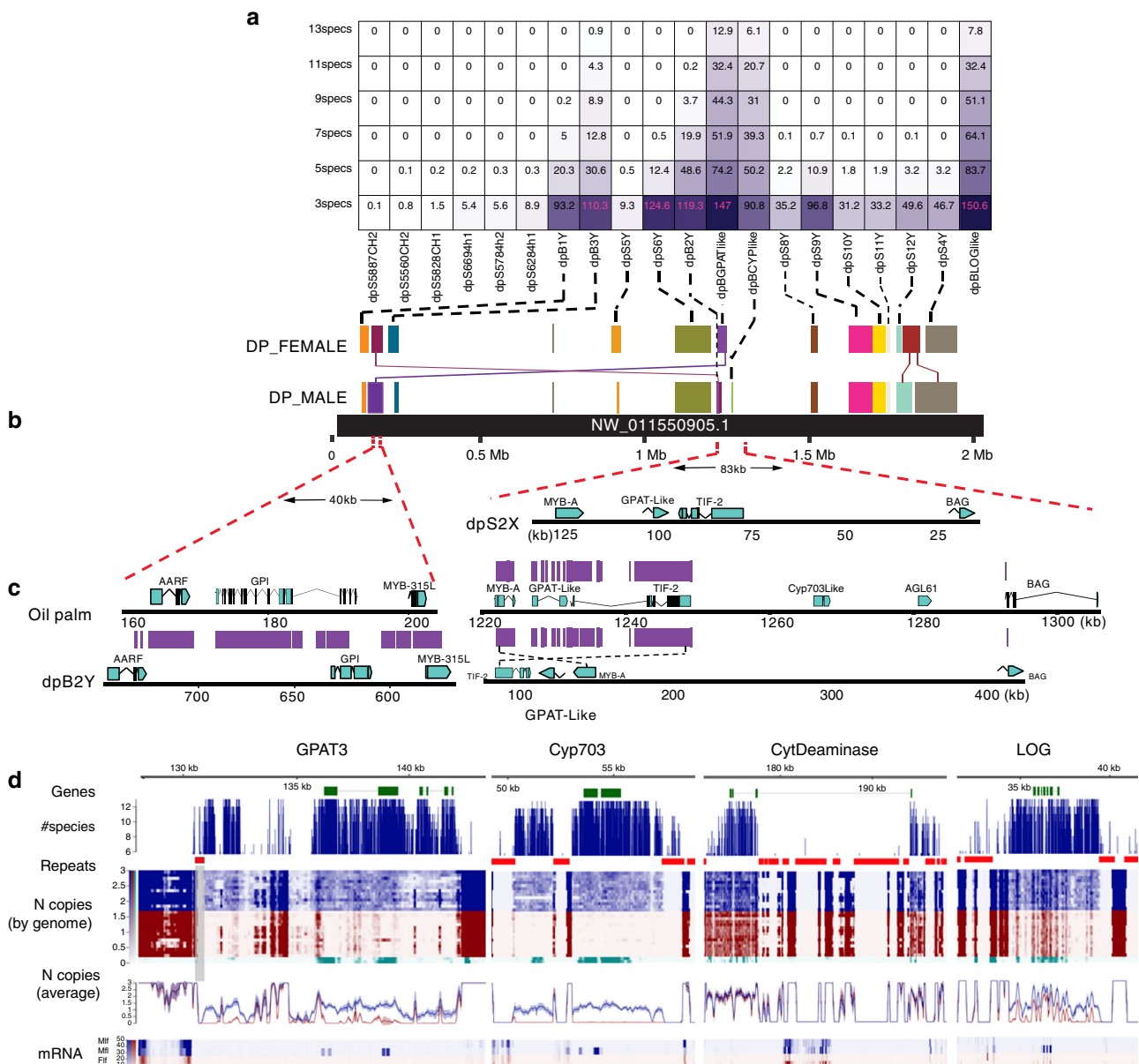

**Fig. 3** The sex determination region in *Phoenix*. **a** *Phoenix* male-specific kmers/bp in date palm scaffolds based on number of species compared (specs: species). Darker boxes indicate higher densities of kmers/bp. Scaffolds with male-specific kmers had similarity to a single locus in oil palm. The only gene not deriving from the region was LOG-like. **b** Sequencing of representatives of both male (DP_MALE) and female (DP_FEMALE) alleles in date palm showed coverage of significant sections of the oil palm scaffold NW_011550905.1. Each date palm scaffold is connected by a dotted line to a colored rectangle showing span of similarity with oil palm. **c** Comparison of the date palm X and Y alleles to the orthologous oil palm region revealed a deletion of the GPAT3 and CYP703 gene in the X chromosome. A similar GPAT3 deletion is found in the Y chromosome although the GPAT3 gene does exist elsewhere on Y. The region showed a fusion/inversion (red dotted lines) with respect to the oil palm locus joining the region downstream of MYB315L to the region upstream of BAG-like in date palm. Note shift of sequence coordinates in dpB2Y with respect to oil palm reveals inversion. Cytidine deaminase resides in this region in oil palm. **d** Analysis of male-specific kmers present in all species of *Phoenix* identified only four scaffolds dpBGPATlike containing GPAT3, dpBCYPlike containing CYP703, dpB3Y containing cytidine deaminase like (CytDeaminase), and dpBLOGlike containing LOG. The most widely conserved kmers were focused in gene and apparently gene regulatory regions (#species track). Sequence coverage of each genome was normalized to 2N and plotted as a heatmap. Blue rows represent males; red rows represent females from the various Phoenix species. Green rows represent hermaphrodite palms Brahea and Livistona. Males showed one copy and hermaphrodite palms showed two copies per genome in most exonic regions (*N* copies, by genome track) with some variation (see text). The average coverage for all males indicated these regions were likely at 1*N* except for the LOG gene that has an autosomal paralog and cytidine deaminase that has an X-linked copy (*N* copies, average track). Gene expression (mRNA track) showed that GPAT3, CYP703, and LOG were all expressed in male flower (Mfl) and was either very low or undetectable in male leaf (Mlf), female flower (Ffl), and female leaf (Flf)

Cytidine deaminase was the only gene with genus-wide male-specific kmers for which both X and Y copies were identified. Phylogenetic analysis of this gene indicates that the male and female alleles form separate clusters, suggesting that their sequences diverged before the speciation events (although the support for clustering was strong only for the X and Y alleles from *P. dactylifera*, *P. atlantica*, and *P. theophrasti*, see Supplementary Fig. 5). One species, *P. rupicola*, was an exception, where we detected large numbers of male-specific kmers surrounding this species' exons, but not within them, so the coding sequences of the X and Y alleles could not be distinguished.

**Synteny with oil palm**. Most genes identified in the male-specific date palm BAC contigs had sequences similar to a single oil palm scaffold (NCBI refseq nucleotide ID NW_011550905.1) (Fig. 3a, b). Furthermore, BAC contigs and sequence scaffolds containing kmers present in many but not in all males of the genus also showed similarity to this oil palm scaffold, and the same gene order was largely maintained between oil and date palm (Fig. 3b, c). The two date palm BAC contigs with the highest density of male-specific kmers contained *CYP703* and *GPAT3*. These genes are in close proximity in the oil palm but the syntenic region in both the X and Y date palm haplotypes lacks both (Fig. 3c), suggesting that recombination at this location occurred even after sex-specific changes. Despite the apparent deletion of these genes from their ancestral location, full-length copies of both *CYP703* and *GPAT3* were nevertheless detected in all males only (Fig. 3d); these genes' location therefore differs between the date and oil palm. The BAC contigs containing the two genes were too repetitive to extend, so joining was not possible. However, they may still be adjacent to each other at the new chromosomal location. A truncated form of the *GPAT3* sequence was found in the region syntenic with oil palm, with part of the 3′ end present in both X and Y alleles. Furthermore, the male-specific BAC contig containing the full-length *GPAT3* gene showed similar truncated forms of the gene just downstream of the full-length copy, so the *GPAT3* gene copy number may have been unstable before the gene became sex-linked.

Our extension of the date palm male allele sequence (dpB2Y) surrounding the *CYP703* and *GPAT3* deletion revealed that the region was fused to a location syntenic to the beginning of the oil palm contig. The oil palm region homologous with the fusion breakpoint is located between a Myb315-like and a cytidine deaminase-like gene, but the date palm Y-linked cytidine deaminase has moved from this syntenic region (Fig. 3), and we found it in one of the repeat-rich male-specific BAC contigs (dpB3Y) that could not be extended to join any other contigs. The Myb315-like gene at the fusion breakpoint was previously identified[15] as a Y-linked gene in the genus *Phoenix* and studied[18] in depth. It is not clear whether this fusion breakpoint is present in the X-linked allele. However, the fact that the cytidine deaminase contains sequences found in all *Phoenix* males investigated suggests that the rearrangement (fusion) of the region with respect to the ancestral oil palm may have played a role in the formation of the Y chromosome though this remains to be investigated. In addition, an inversion with respect to date palm BAG-like gene was noted in the male allele only (Fig. 3c). The only male-linked sequence that did not show synteny to the oil palm scaffold was the BAC contig containing the LOG-like gene described above (Fig. 3).

**Inference of natural selection**. Analysis of the synonymous substitution rate ($d_S$) in each of the four candidate genes in the lineages from the common ancestor to the 13 *Phoenix* species

males (using phylogenetic trees rooted with the respective oil palm) revealed different divergence rates among the genes. Across all species, *CYP703* had a mean $d_S$ of 0.30 with standard deviation (S.D.) of 0.01, and cytidine deaminase had a similar rate with a mean $d_S$ of 0.28 and S.D. of 0.02. These were followed by *LOG* (mean $d_S$ 0.21, S.D. 0.01) and *GPAT3* (mean $d_S$ 0.20, S.D. 0.01). In comparison, a set of 22 genes conserved in all sequenced plant genomes termed universal single copy orthologs[24] yielded a mean $d_S$ of 0.14 and S.D. of 0.03. Possible reasons for the variation in rate of synonymous substitution in the male-specific genes versus the other genes are explored in Discussion.

Using the inferred genealogies from alignments of the sequences of the four genes across the *Phoenix* species and the two monoecious palm species, we inferred the relative times to the most recent common ancestor for each gene in the *Phoenix* species. As noted above, all four of the genes have a gene genealogy that is concordant with a monophyletic origin across the genus. Using TimeTree[25] we inferred the time in the past when each gene tree converged, indicating the time of origin of each gene as a proportion of the time back to common ancestry of the monoecious and dioecious palms. For CYP703, GPAT3, and LOG, these values were 13.4%, 15.2%, and 9.2%, suggesting that the LOG gene may have become Y-linked more recently than the other two (see Discussion below). Cytidine deaminase could not be analyzed in this way because the monoecious species' sequence falls within the date palm clade, suggesting that there might have been some X–Y gene conversion or other complication.

To assess whether the Y-linked genes in the genus *Phoenix* have evolved under selective constraint, we tested for neutrality by comparing between-species divergence at synonymous and nonsynonymous sites ($d_S$ and $d_N$ values, respectively). The analyses detect purifying selection, with $d_N/d_S < 1$. Likelihood ratio tests confirm the significance of reduced rates of substitution at nonsynonymous sites for all genes, and the four values of mean $d_N/d_S$ across the *Phoenix* species are: CYP703 (0.16), GPAT3 (0.25), LOG (0.12), and Y-linked copies of cytidine deaminase (0.16). In addition, codon-based tests in HyPhy[26] inferred many codons in each gene showing strong purifying selection. While it is possible that the genomic rearrangements described above could create adaptive pressures that might drive positive selection on the sex-specific genes, this analysis suggests purifying selection, suggesting conserved functions throughout the radiation of the *Phoenix* species.

**Gene expression analysis**. Analysis of RNA from male and female leaves and flowers showed that *CYP703* and *GPAT3* are expressed only in male flowers (Fig. 3d). Cytidine deaminase, whether from X or Y alleles, did not show significant expression in any of the four tissues investigated. The *LOG* gene is highly expressed in male flowers and more weakly in female flowers, consistent with a possible role in suppressing female flower formation in males. However, as mentioned, the exons of *LOG* contained no male-specific kmers. SNP analysis showed that the reads in our expression analysis likely derived from the autosomal copy of this gene. Expression in male flowers appeared highest for the five 3′ exons, whereas expression in female flowers was evenly low across the gene body. Whether this reflects functional differences will require further investigation.

## Discussion

Localizing sex determination genes and understanding their roles in sex chromosome evolution is challenging if recombination is suppressed between the sex chromosomes. Our approach identified a small number of genes present only in the males of the dioecious *Phoenix* species (Table 2). Divergence between X and Y

**Table 2 Genes identified in this study**

| Gene | Putative ortholog other species | Presence in date palm female reference genomes or transcriptome | Function in other plants | Evidence for purifying selection[a] | Interpretation |
|---|---|---|---|---|---|
| A Cytochrome P450 (CYP) | Rice CYP703A3 | No | Male reproductive organs and male fertility | Yes, $\omega = 0.16$ | Recessive male-sterility mutations in one of these genes were the initial step producing females, or deletions (alternatively, deletion of both genes followed loss of male fertility due to mutation in one gene or the other). A chromosome rearrangement moved the genes to a new location |
| Glycerol-3-phosphate acyltransferase 3-like (GPAT3-like) | Rice or Arabidopsis GPAT3 | No | Male reproductive organs and male fertility | No, $\omega = 0.25$ | Same as above |
| Lonely-Guy (LOG) | LOC105055182, on a non-homologous chromosome, 12, in oil palm | No | Female flower development | No, $\omega = 0.12$ | Duplication (translocation) of paralogous autosomal gene (date palm Pd_LOC103701078) after divergence of Elaeis and Phoenix. The translocation into the proto-Y non-recombining region led to the suppression of female flowers in males, i.e. to dominant female sterility |
| Cytidine deaminase-like | LOC105059743 in oil palm (guanine deaminase-like) | Yes | No clear floral function | No?, $\omega = 0.16$ | Possibly at boundary of inversion, no clear functional interpretation |

[a] $\omega = dN/dS$ or nonsynonymous to synonymous mutation rate ratio

extends over multiple megabases[15,16], but the main region of divergence between X and Y includes a genome region in which two genes appear to have been deleted from females and relocated in males. Strikingly, deletion of these genes in other monocots results in male sterility. Importantly, three of the male-specific sequences, *CYP703*, *GPAT3*, and *LOG*, appear to be single copy in males, which excludes the possibility that these sequences could simply have been duplicated sequences into a non-recombining Y-linked region. Our approach requires that divergence among all the species in a genus is sufficiently low that sex-specific genes still have kmers shared between the species. Genes with essential functions in males might then be the only ones retained in all the species, allowing one to identify genes with functions in sex determination. Indeed, TBLASTN analysis of the male BAC contigs identified one other gene, MAP1, located downstream of cytidine deaminase which contains no male-specific kmers, suggesting it may have been lost from or degraded on the Y, except in the date palm.

Our identification of *CYP703*, *GPAT3*, and *LOG* genes is encouraging, because the first two have functions in male reproductive organs and male fertility, while *LOG* genes are associated with female flower development. The role of these genes in *Phoenix* remains to be investigated. However studies of the likely orthologs in other monocots are interesting. Mutants of *CYP703* in rice[22] and maize[27] and *GPAT3* in rice[23] reveal function in both pollen formation and/or anther development through their indispensable role in various lipid synthesis pathways. In rice, *GPAT3* and *CYP703A* are expressed in tapetal cells[23,28] which are responsible for synthesis and secretion of sporopollenin precursors, a major component of the outer pollen cell wall[29].

*CYP703* is a single member gene family found across land plant taxa, suggesting that it encodes an essential function[30]. Knockouts of *CYP703* homologs in Arabidopsis[21], rice[22], and maize[27] resulted in male sterility with no apparent effect on vegetative growth. As a single member gene family, a paralog in *Phoenix* that could compensate for a deletion likely does not exist.

Similarly, deletion of *GPAT3* in rice resulted in complete male sterility without affecting normal vegetative development[23], and the sterility was recessive. A single copy Y-linked *GPAT3* in *Phoenix* could thus sustain normal male flower development. In rice *gpat3* mutants, most pollen grains had abnormal pollen wall development, never fully matured and were later aborted[23], similar to rice *CYP703A* mutants. Expression of *CYP703A3* was significantly reduced during flower development in these mutants, suggesting a possible functional connection between these two genes.

The *LOG* or LONELY GUY family of genes is involved in activation of cytokinin, an important phytohormone. Functional analysis of the first identified *LOG* member in rice showed that all rice *LOG* mutants were impaired in normal flower development, producing flowers that lacked ovules[31] or flowers with no pistil and a single stamen[32]. Early inflorescence development in these mutants was normal, but floral meristems were later aborted, a phenotype similar to date palm, where both flower sex organs are initiated in early flower development[33].

Any sequence in the Y-linked region may evolve male-specific kmers without being involved in sex determination. It is, however, intriguing that only one sequence with conserved kmers in all males investigated has no clear floral function; this is the cytidine deaminase-like gene, which is located at the border of the fusion between sequences surrounding the *Myb315*-like and *BAG* genes. Both X- and Y-linked regions retain copies of the gene, although it is unclear if both are functional. It is possible that cytidine deaminase is simply a passenger closely linked to the sex-linked chromosomal inversion described above (Fig. 3c), and its role in sex determination, if any, remains to be investigated.

We therefore suggest that the genus *Phoenix* ancestor may have evolved from hermaphroditism to dioecy through a gynodioecious intermediate. Gynodioecy would have occurred through loss of male function by recessive male-sterility mutations (as proposed by Charlesworth and Charlesworth[3]) either involving or followed by deletion from a proto-X chromosome of *CYP703*

and/or *GPAT3* which left both genes intact on the proto-Y (Fig. 4a). That CYP703 acquired its Y-linked status early is supported by both the $d_S$ estimates and the TimeTree analysis. The rearrangement that moved *CYP703* from the region sytenic with oil palm to its current location on the Y chromosome is probably the same event that also brought *GPAT3* and cytidine deaminase (Fig. 4b). Finally, we propose that the *LOG* gene translocated into the proto-Y non-recombining region with the putative role as a suppressor of female flowers in males, possibly via expression of a truncated LOG competitor (Fig. 4c). The TimeTree dating suggests that the *LOG* gene became part of the non-recombining region later, and the slightly more neutral pattern of divergence of the Y-linked LOG sequence ($d_N/d_S = 0.21$) might be consistent with a negative action, which might create only slight selective constraints on the sequence. Although the *LOG* gene is the only male-specific gene whose family function is known to be important for female flower function in other plants, its function in *Phoenix* will require further research.

Altogether, our results suggest that *Phoenix* sequestered some key genes necessary for male flowers on the Y chromosome, and likely suppressors of female flowers as well. Our results support the theory that dioecy developed prior to the species radiation in *Phoenix*. Of high interest is that male-specific sequences maintained across the genus include only four genes, three of which are known to be important in male and female flower function in other monocots. Whether these gene sequences could allow for dioecy to be introduced into close relatives such as oil palm will be of biotechnological interest in the future.

## Methods

**Sample collection and genome sequencing.** Leaves of *Phoenix* species were collected by taxonomic and palm specialists in the field at various locations including the USDA in California, the Universidad Miguel Hernandez in Alicante, Spain, and the Huntington Gardens in California (Table 1). Leaves from multiple individuals from all 14 *Phoenix* species[34] were collected and one male and one female individual of each species was sequenced and analyzed in this study (with the exception of *P. pusilla* for which a true to type male could not be found). Leaves from the monoecious palm species *Livistona rotundifolia* and *Brahea dulcis* were included for control. DNA was extracted from leaf material using the DNeasy Plant Kit (Qiagen). DNA sequencing was conducted on the Illumina 2500 and 4000 according to the manufacturer's recommended paired-end sequencing protocol with libraries of ~400 bp. Sequence coverage was determined by analysis of the most prominent kmer peak using JELLYFISH[35] and by alignment to the date palm genome followed by empirical inspection of coverage at 5000 SNP sites in three scaffolds not showing linkage to sex determination.

**Kmer analysis.** We attempted to find kmers of defined sequence length conserved in all members of one sex yet absent in all members of the other. First, to determine the kmer length of appropriate sensitivity and specificity, we identified the maximum length of kmer that was likely to be conserved in all members of the genus despite normal sequence divergence. To that end, sequencing reads from all species were aligned to the date palm reference (NCBI accession ACYX00000000) using BOWTIE2 (ref. [36]) and SNPs were called using SAMTOOLS[37] while avoiding annotated repeats and positions with excessive coverage (greater than 3X the mean coverage). Four scaffolds with either no male-associated SNPs in more than six species were utilized for analysis (dpS5505H2, dpS5560CH1, dpS5784H1, dpS5828CH2) spanning a total of 2,870,261 bp. Non-repetitive sequence of 1,780,052 bp from the selected scaffolds contained 107,692 SNPs identified in at least one of the 13 male genomes. This revealed a rate of approximately one polymorphism per 16.5 bp among all male members of the species tested. Distances between polymorphisms in the genus were not normally distributed with SNPs more likely to cluster near each other. Indeed, 68% of SNPs occurred within 16 bp of each other. However, scanning the test regions in 500 bp windows with a step size of 250 bp we did not identify any of the 11,483 windows that had less than 20 sites (40 including reverse complement) of 16 bp fully conserved across all males. Therefore selecting a kmer size of 16 bp had an empirical probability of less than $8.7 \times 10^{-5}$ of not identifying a 500 bp or greater fragment of DNA that was unique to males in the genus.

To identify sex-specific kmers, 16 bp kmers were extracted from the raw sequencing read (FASTQ) files using JELLYFISH[35] for each genome. A genome-specific cutoff was selected based on coverage (ranging from 3 to 15) under which kmers were assumed to derive from errors in reads and were discarded. Files of

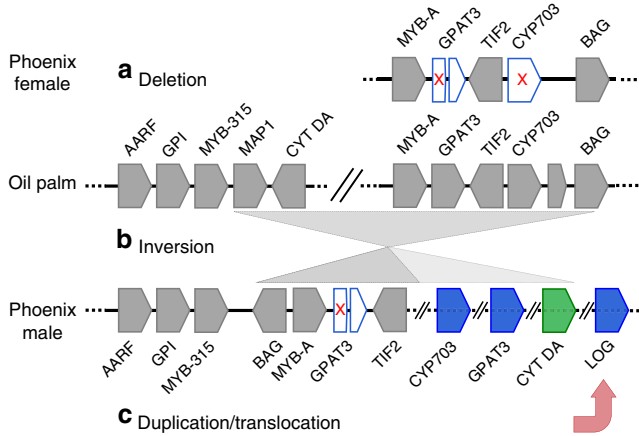

**Fig. 4** A model for the development of dioecy in the genus *Phoenix*. The genes in the region as found in female scaffold dpS2X, oil palm scaffold NW_011550905.1, and male contig dpB2Y. The fact that the male-specific sequences are focused in this region suggests that it may be the origin of sex determination in *Phoenix*. **a** The first step would be the deletion (white pentagons with red x) of *GPAT3* and/or *CYP703*, two genes known to be critical to male flower formation, leading to gynodioecy and formation of a proto-X chromosome. **b** An inversion (gray triangles) of the corresponding region on the normal chromosome moving the cytidine deaminase (CYT DA) gene would create a proto-Y chromosome with recombination arrest. **c** This would be followed by a duplication and translocation (red arrow) into the region of the *LOG* gene creating the final Y chromosome with suppression of female flower development. NCBI GENE IDs as follows AARF: LOC105059738, GPI: LOC105059739, MY315L: LOC105059740, MAP-1: LOC105059742, CYT DA: LOC105059743, MYB-A: LOC105059783, GPAT3: LOC105059961, GPAT3tr: truncated GPAT3, TIF2: LOC105059784, CYP703: LOC105059962, BAG: LOC105059785. The intervening gene between CYP703 and BAG was not present in either male or female alleles of date palm. A short, truncated 3′ end of GPAT3 remains in both X and Y alleles (white pentagon with no red x)

male- or female-specific kmers were created for each species by removal of kmers matching those in the opposite sex of that species.

Sequencing reads containing kmers present in 13 males and absent from all 14 females were selected from the genome FASTQ file of the Deglet Noor BC5 male. These were assembled to create short contigs on which PCR primers were designed to screen a date palm BAC library. Male-specific sequences were then validated by checking for their presence in an additional 13 males or females from various species of *Phoenix* (Supplementary Table 1).

**Sequence assembly and annotation of X and Y alleles.** Two BAC libraries (Amplicon Express Inc., Pullman, WA, USA) were constructed from fresh leaves of *Phoenix dactylifera* Deglet Noor BC5 male to a predicted coverage of 6X each for a total of 12X genome. Libraries were constructed in the pCC1 BAC cloning vector with an average insert size of 125 kb using a partial digest of *Hind*III or *Eco*RI.

PCR primers (Supplementary Table 3) were designed in regions of the date palm genome previously identified as linked to sex[15] to probe the date palm male BAC library according to the manufacturer's recommended protocol (Amplicon Express Inc., Pullman, WA, USA)[38]. DNA from BAC clones identified by PCR screening was extracted from overnight *E. coli* cell cultures using the Large-Construct kit (Qiagen). Libraries were constructed and sequenced on the Pacific Bioscience RSII sequencer according to the manufacturer's recommended protocol for de novo sequencing (Pacific Biosciences, Melon Park, CA). Libraries were size-selected from 10 to 40 kb using BluePippin (Sage Science, Beverly, MA). Only reads longer than 10 kb were utilized to assemble individual BACs utilizing the HGAP assembler (Pacific Biosciences, Melon Park, CA). Effectively all BACs assembled into a single contig. Consensus sequences of multiple BACs were obtained using Gap5 v1.2.14 (ref. [39]).

Sequenced BAC clones were checked for the presence of X- or Y-allele-specific polymorphisms. Where possible, a clone from each allele was selected for extension by further BAC library probing and sequencing.

For phased allele sequencing, the Deglet Noor BC5 (DNBC5) genome was subjected to 10X Genomics (Pleasanton, CA) library construction and sequencing per the manufacturer's protocol. Data were output with the 'pseudohap2' model

that attempts to separate haplotypes during sequence assembly. Male-specific kmers found in the DNBC5 genome and at least two other species' males but not their counterpart females were used to search the phased haplotype scaffolds for X and Y linked scaffolds. Scaffolds with at least 100 male-specific kmers and six times more male-specific kmers in one haplotype were selected for further analysis (Supplementary Table 2).

For annotation, REPEATMODELER was used to identify new repeats followed by REPEATMASKER[40] to mask repeats in BAC contigs and 10X scaffold sequences. Gene predictions were obtained using a combination of the programs FGENESH++ (Softberry Inc., USA), Augustus[41], SNAP[42], and manually curated gene models. The putative role of the candidate genes was determined by BLAST searches against the NCBI database and the oil palm annotated reference genome ASJS00000000 (ref. [43]).

**RNA-seq and gene expression analysis**. Date palm flowers and leaves were collected from two separate male and two separate female trees at anthesis during the flowering season of 2016 in Doha, Qatar. Total RNA was extracted according to the protocol described by Chang et al.[44] followed by on-column DNase treatment using the RNeasy plant mini kit (Qiagen). mRNA-seq libraries were constructed using the Ovation RNA-seq V2 kit (NuGen, San Carlos, CA) according to the manufacturer's protocol. Libraries were sequenced using either 75 or 150 bp read-lengths on a HiSeq4000 (Illumina, CA) according to the manufacturer's recommended protocol. Sequences were quality trimmed, replicates combined, and aligned to repeat-masked scaffolds using HISAT2 (ref. [45]). Sequence counts were normalized to fragments mapping per kb of non-sex-linked control scaffold sequence.

**Phylogenetic analysis**. Phylogenetic analysis of genes was conducted on de novo assembled sequences for each species where possible. Briefly, for each species' male, genome sequence reads matching the male-specific kmers were collected with their paired-end mates and assembled using the SPADES assembler[46]. Where possible, the same approach was used on the samples from the hermaphroditic genus *Brahea* and *Livistona*. Consensus sequences for each species were then trimmed to allow for global alignments. For cytidine deaminase, male-specific kmers were too few to collect enough sequence for routine assembly. Therefore, the female reference sequence was used and SNP positions substituted with male-specific alleles. Multiple sequence alignments and phylogenies were inferred using MEGA7 with default parameters[47]. Coding sequences were aligned with ClustalW and phylogenies were inferred by Maximum Likelihood using the Jukes-Cantor model based on 1000 bootstraps. Synonymous ($d_S$) and nonsynonymous ($d_N$) substitution rates were calculated based on the Nei-Gojobori method implemented in SNAP v2.1.1 (ref. [48]). A set of 22 genes conserved in all plant genomes termed universal single copy orthologs[24] were selected from date palm and oil palm for understanding average synonymous substitution rates between the two species.

## Data availability

All contigs/scaffolds were deposited in Genbank with Accession numbers MH680964 to MH681004 (Supplementary Table 2). Sequences used for phylogenetic analysis including newly identified male genes from the various species were deposited in Genbank with Accession numbers MH668887 to MH668902, MH668903 to MH668917 and MH685629 to MH685642. All species' whole-genome sequencing reads have been deposited in the Sequence Read Archive (SRA) at NCBI under accession code SRP128017 and the Bioproject under accession code PRJNA427409. RNA-seq data were deposited at the SRA under the study number SRP148811.

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

## Acknowledgements

We thank Sean Lahmeyer at the Huntington Gardens for his kind assistance with collection of palms for this study. We thank Encarnacion Carreño from the University of Murcia and Concepcion Obón from the University of Miguel Hernandez (National *Phoenix* Palm Germplasm Repository of Spain) for their assistance in collection of *Phoenix* species. This study was made possible by grant NPRP-EP X-014-4-001 from the Qatar National Research Fund (a member of Qatar Foundation).

## Author contributions

M.F.T. conducted the BAC library probing, RNA-seq libraries, assembly, phylogenetic analysis, and wrote the manuscript; L.S.M. conducted genome sequencing, library construction, and helped write the manuscript; I.A. conducted bioinformatics analysis and helped write the manuscript; I.K.A.A conducted RNA-seq libraries and sequencing; R.K. and D.R. maintained palm collections, provided phenotyping and systematics analysis; Y. A.M. directed in library construction and sequencing and conducted bioinformatics analysis; A.G.C. conducted evolutionary analysis and helped write the manuscript; K.S. conducted bioinformatics analysis and helped write the manuscript; J.A.M. envisioned the project, conducted bioinformatics analysis, and wrote the manuscript.

## Additional information

**Competing interests:** The authors declare no competing interests.

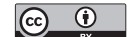

