## [Peer Review File · Nature Communications]

Reviewers' comments:

Reviewer #1 (Remarks to the Author):

My review is attached, along with an edited version of the Word file, and a suggested table.

Reviewer #2 (Remarks to the Author):

Sex determination is a major switch in the evolution of dioecious plants and animals. It used to be a major effort and takes many years to identify candidate genes for sex determination. With the rapid advancement of genomic technology, the authors sequenced the male and female genomes of 14 species in the genus *Phoenix* and identified 4 shared male specific genes. Among the four genes, CYP703 and GPAT3 are known to cause male sterility in monocots, and the LOG-like gene is associated with female flower development. Deletions of CYP703 and GPAT3 in X chromosomes are detected across all 14 species, and one deletion would be sufficient to cause male sterility to initiate the transition from hermaphrodite to gynodioecy. The LOG-like gene was translocated into the non-recombining region of the date palm Y chromosome. The authors' interpretation of a truncated LOG expression silenced the normal function of LOG to suppress carpel development is plausible. The purifying selection of CYP703 and GPAT3 and near neutral divergence of LOG-like gene among *Phoenix* species support their interpretation. This innovative genomic approach is suitable for moderately ancient sex chromosomes that are ancestral at the genus level, allowing substantial degeneration of the Y chromosome in each species, resulting a small number of shared Y-specific genes across the genus. It might be suitable in young sex chromosomes that were evolved at the species level but shared the same sex determination genes. The indispensable role of CYP703 and GPAT3 in various lipid synthesis pathways leading to a new role of sex determination is a novel finding, emphasizing the importance of gene network for understanding underlining mechanisms of various biological morphologies and functions. Their work is an excellent contribution to the sex chromosome research community, and will be of interest to a broad group of researchers.

A minor change:

The clustering of all X or Y alleles across 14 or 13 species is a major piece of evidence to conclude that sex chromosomes are ancestral in the genus Phoenix. For this reason, I would suggest moving sup figure 5 to the main text.

Reviewer #3 (Remarks to the Author):

The manuscript "Genus-wide sequencing supports a two-locus model for sex-determination in Phoenix" includes some interesting data and analyses aimed at understanding the origin and evolution of X and Y sex chromosomes in the date palm genus, Phoenix. The findings, however, are presented in way that is difficult to follow and the conclusions are FAR too speculative. In my opinion the manuscript must be overhauled in order to make the results more accessible and speculations should be reframed as hypotheses to be tested through functional analyses of gene function.

Some general comments and suggestions for improvement include:

1. Do not forget to acknowledge the work of Westergaard (e.g. 1958, Adv Genet) in the introduction and discussion. This work was really the basis for the more formal models developed by Charlesworth & Charlesworth.
2. The results of the comparative K-mer analyses (Figure 1a) are quite impressive and key for identifying the putatively sex-linked BACs and 10X-Genomics scaffolds, but the in my opinion the manuscript belabors the K-mer analysis results. The work could be presented as building on the K-mer analysis results as follows: 1) K-mer analyses implicate Y-specific sequences; 2) Assemblies of reads with Y-specific K-mers used to develop probes for BAC libraries and 10X-Genomics scaffolds; 3) Consensus sequences assembled for Y and X haplotypes for the putatively non-recombining sex determination region of the date palm genome (Figure 3); 4) Annotations of these consensus sequences reveal putative Y-specific genes and structural differences between non-recombining portion(s) of the Y and X chromosomes in date palm; 5) Annotations and phylogenetic analyses of these genes suggest the timing of duplication and loss events relative to the split between oil palm and date palm lineages and origin and early diversification of Phoenix species; 6) Gene homologies provide basis for HYPOTHESIZED gene functions.
3. Most of the gene trees are not well resolved. Are they base on the coding sequence alone? If so, could adjacent non-coding sequence be aligned among Phoenix species and included in the analyses?

4. I did not see characterization of scaffold lengths and the quality of the 10X-genomics assembly. It seems these were important for identifying and characterizing the X and Y haplotypes, so the 10X-Genomics assembly should be described in detail.

5. How large in the non-recombining portion(s) of the Phoenix sex chromosomes? Do the assemblies completely span this region? Can recombining segments be identified as the ends of the consensus X and Y assemblies? Are the authors certain they have identified all of the male-specific genes? How about the non-coding regulatory RNAs (e.g. as described for sex determination in persimmon)?

6. How does the data presented in the manuscript allow any inference about the evolution of gynodioecy? Without any data on gene function, how can the authors assert that their data implicate a two-gene sex determination system. How can the authors reject the possibility that one of the inferred Y-specific genes is acting as a master regulator directing gene interactions that suppress female organ development and promote male function.

In general, the manuscript is poorly organized and many conclusions are not well supported. At the same time, the work does advance understanding of the Y and X chromosome gene content and structure in date palm.

COMMENTS FOR AUTHORS

This ms sounded very interesting, based on the abstract, which suggests that the study establishes that the sex-determining region of these plants carries distinct male- and female-sterility mutations, as has been hypothesized. The evidence for this hypothesis has previously been purely genetical, but it is interesting to identify the genes and see whether their properties correspond with the evolutionary hypothesis. The ms does contain some interesting results, and they do appear to support the hypothesis, but the evidence is presented in a strange and unhelpful manner, that leaves important information until the Discussion section, so that, as I was reading the results part my mind was occupied with an alternative possibility that is not discussed — that the male-specific gene sequences detected by the study might reflect common ancestry of the non-recombining sex-determining region of these plants, and not necessarily genes that underwent the male- and female-sterility mutations that were functionally involved in the evolution of separate sexes. This possibility needs to be explicitly mentioned and excluded. The available information can be marshalled to argue against it, but it should be made clear and examined explicitly. I believe that this can easily be done by arranging the material much more clearly, and adding some further information, and some simple extra analyses of the sequences.

The approach used sounds clever — knowing that these Phoenix palm species have male heterogamety, the study searched for a possible Y-linked region by sequencing males and females of several species to see if they can find sequences that are present only in males. This indeed yielded 1653 16bp kmer sequences present in males of all 13 species where males were studied, and absent in females of all the species, plus another species where no male was obtained. (The sample sizes can be explained more clearly than in the submitted ms; also the statement that “kmers present in females and absent in males were not present in more than 8 species of the genus” is not very clear. I understand that no such sequence was found in females of all species, but the number of kmers should be mentioned, to compare with the number of male-specific ones — presumably it is a smaller number).

The sequences were used to identify BAC clones and thus to identify candidate sex-determining genes. This revealed only a few genes common to males of all species (or maybe just to 6 of them shown in Figure 2) surrounded by highly repetitive sequence. I did not understand Figure 2. The legend mentions several genes (GPAT3, CYP703, LOG and cytidine deaminase or dpB3Y contig) that appear not to be shown in the figure, and the phrase “for comparison see synonymous mutation rates in the same genes” is obscure — are those mutation rates (or perhaps substitution rates are meant) shown somewhere in the ms?

It would seem important to analyze these BACs to try and discover all the genes present in them, not just male-specific ones. As presented, it is unclear to me whether several, or even many, other genes are present. Reference number 15 cited by the authors mentions that the “region segregating with gender [presumably in male meiosis] is approximately 26 cM in length” [presumably, in female meiosis],

indicating a total physical length possibly as long as 13 Mb or 2% of the plant's genome". Certainly a 26 centiMorgan region suggests a physically large region that might include many genes, and I think that readers need to know what proportion of the region has probably been sequenced in this study, and how much remains unexplored, and how many genes might be included in the unexplored regions.

The apparently low gene density, based on the BAC sequencing, is consistent with either the region in question having been non-recombining for a long time before dioecy evolved (e.g. a centromeric region), or with enough evolutionary time since Y-linkage evolved that repetitive sequences have accumulated. Under the second possibility, sequence divergence between the X- and Y-linked regions should be large enough that it can be estimated using the coding regions and introns of any genes that are present in both haplotypes, and the divergence ought to be less than between the X-linked sequences and the orthologs in the outgroup species, before dioecy evolved. It would be good to present such results. If there really are no genes other than the small number identified by the search for male-specific sequences, this should be explained, so that readers understand why this analysis was not done (although one gene with copies in both the X- and Y-linked regions was identified, and might be informative). I don't think that there was a complete absence of other genes, as page 9 mentions that "TBLASTN analysis of the male BAC contigs identified sequences such as MAF1 downstream of cytidine deaminase, having sequence similarity to known genes but no male-specific kmers. These genes are likely not critical to dioecy as they have degraded faster among species". And page 10 mentions a *BAG* gene.

I also feel that the four male-specific genes should be more clearly explained. A table would be helpful. The section of the text describing analyses of the male-specific genes that were identified (page 5 says "highly conserved male-specific genes" and doesn't explain why all 4 genes were not examined). Later in the text, a similar restriction appears, but in other places all 4 genes are discussed. To try and understand the results, I summarized them for myself and I attach the crude table.

The case that best supports the idea that a gene has been lost from the X-linked region (potentially causing females' male-sterility) is the CYP gene the study identified. Similar CYP703 family sequences were found in 11 other plant species, with multiple gene family members presumably found in at least some of them, since 32 proteins were identified. There was no sign of complete or partial sequences of *CYP703* in the published date palm female reference genomes or transcriptome data. But this is far from establishing that this gene really is missing from females (admittedly not an easy task), or that its loss led to male-sterility.

Assuming that males of all species (and not females) carry the genes identified, the question is how this can be interpreted. If these plants have a physically large region that stopped recombining in their common ancestor, then, given enough evolutionary time, different species' Y-linked regions will have diverged from one another, both in sequence and in their arrangement and gene content, as has occurred in the Y-linked regions of mammalian genomes, and even among different primate species. If this is the situation, then selecting just those sequences that are

universally male-specific should identify genes that have been retained because of their essential functions in males, including presumably the male-determining factor. It would be helpful to explain the basic reasoning and the principle of the test employed, which is not explained in the present version (making this clear at an early stage would also remove the need for the first part of the final section). Complete absence of a gene from female palms would seem surprising, but the text does not outline what might lead to such a situation.

Before we can conclude that the study identifies “the changes foundational to dioecy in *Phoenix*”, we should also exclude other possibilities. A Y-linked region that evolved in the common ancestor of a group of related species might have undergone changes that are not related to involvement in male function. The phrase used on p. 10 (“a passenger of the process of sex chromosome evolution”) presumably refers to some such idea, but in my opinion this is too vague, and the thought should be made clear and explicit. A sequence that is absent in females could, for instance, reflect a duplication of a gene, followed by sequence divergence. The sequence might still be present in the females’ X-linked region and also potentially elsewhere in the Y-linked region, but divergence could make these copies hard to detect. The results indeed suggest that some genes lack X-linked copies, suggesting that translocations have occurred into the Y-linked region (as the ms itself mentions). Again, such changes might not be part of the initial evolution of separate sexes, as they are thought to appear during the later stages of sex chromosome evolution, e.g. in *Drosophila* species. With a physically large non-recombining region that might be prone to rearrangements, these suggestions are quite plausible.

The evidence against this alternative explanation for the results, and favouring the authors’ interpretation ought (in my opinion) to be described, or at least mentioned, at an early point and not delayed until the Discussion section. This finally appears on p. 9, where the evidence concerning the male-specific genes’ functions in other plants is discussed, which shows that the mutations in the genes identified by this study cause male or female sterility in other plant species. I include this information in the summary table that I mentioned above. If such a table were present in the paper, the authors could draw attention to important ideas by referring to the table, and then the strange organization of the material might be ameliorated, or the table might help the authors arrange their evidence better.

An important component of the two-gene hypothesis is not mentioned, however. This is that the proposed dominant female sterility mutation that defines the Y haplotype (along with absence of the male-sterility mutation) should increase the fitness of males, while decreasing that of females. If a complete female sterility mutation was involved, this clearly has the latter effect. However, it will not invade a gynodioecious population unless male function is better than that of the hermaphrodites that are present. This may be difficult to test, but the need for a test (and the point that the study does not fully test the 2 gene model) should be mentioned, especially as the ms mentions the sort of advantages that allow females caused my male-sterility mutations to spread. Is a male-enhancing effect of LOG

likely?

OTHER COMMENTS

It should be made clear in the Introduction (not the Discussion) that hybridization can occur among different species in the genus, as this conveys some information that the species are quite close relatives. It is also relevant when discussing X-Y divergence between the clades (and within the species, as I suggest should be added).

Page 6 includes analysis of the synonymous mutation rate (dS) in the four genes for all *Phoenix* species compared to the ortholog in oil palm. The oil palm appears not to be explained in the manuscript, so this comparison is difficult to understand. Is this species a non-dioecious (maybe monoecious) outgroup and does it belong to a genus other than *Phoenix* (page 5 mentions *Elaeis*) without explaining this genus? This analysis revealed higher synonymous substitution rates in the sex-specific genes, presumably implying acceleration in the dioecious species. The ms says that possible explanations are in the Discussion, but I could not find them.

The analyses to detect purifying selection are inconclusive. They suggest purifying selection (which would suggest that a gene has retained a function), or perhaps neutrality (which is possible for a gene has undergone a non-functional duplication in the Y-linked region, and is absent from the X). Again, the motivation for the analyses should be outlined.

Page 6 mentions that the inferred proportions of the time back to common ancestry of the monoecious and dioecious palms for convergence of each gene tree is quite small (the highest value was only 13.4%, for CYP703). It is stated that the low values are consistent with their “arrival” on the Y chromosome, but, at this point, the reader is unaware of the interpretation involving a gene translocating to the Y from other initial genome locations. This seems possible (as mentioned above), but it does not relate closely to the 2 gene hypothesis for the evolution of dioecy that the abstract mentions, and it would be helpful to refer to the Discussion section, where this is discussed.

The final paragraph repeats what has been said already, and can be omitted, particularly as some of the text over-states the case. Specifically, it is not correct to say that the “results provide evidence that *Phoenix* passed through a gynodioecious intermediate” — they can be viewed as consistent with that model.

Several references are not the most appropriate. For example, the originator of the two-gene hypothesis is Westergaard (he reviewed the evidence in its support in 1958. The mechanism of sex determination in dioecious plants. *Advances in Genetics* 9: 217-281). Another example is that Spigler et al (reference ³⁵) are cited for the statement that (in my shortened wording) “Females could spread in the population through increased female seed production. The 1978 paper by Charlesworth & Charlesworth deals with this, as does Lloyd’s paper, Lloyd DG 1975. The maintenance of gynodioecy and androdioecy in angiosperms. *Genetica* 45: 325-339.

Response to reviewers' comments for “Genus-wide sequencing supports a two-locus model for sex-determination in Phoenix” - Manuscript NCOMMS-17-24316-T

Reviewer #1 (Remarks to the Author):

My review is attached, along with an edited version of the Word file, and a suggested table.

COMMENTS FOR AUTHORS

This ms sounded very interesting, based on the abstract, which suggests that the study establishes that the sex-determining region of these plants carries distinct male- and female-sterility mutations, as has been hypothesized. The evidence for this hypothesis has previously been purely genetical, but it is interesting to identify the genes and see whether their properties correspond with the evolutionary hypothesis. The ms does contain some interesting results, and they do appear to support the hypothesis, but the evidence is presented in a strange and unhelpful manner, that leaves important information until the Discussion section, so that, as I was reading the results part my mind was occupied with an alternative possibility that is not discussed — that the male-specific gene sequences detected by the study might reflect common ancestry of the non-recombining sex-determining region of these plants, and not necessarily genes that underwent the male- and female- sterility mutations that were functionally involved in the evolution of separate sexes. This possibility needs to be explicitly mentioned and excluded. The available information can be marshalled to argue against it, but it should be made clear and examined explicitly. I believe that this can easily be done by arranging the material much more clearly, and adding some further information, and some simple extra analyses of the sequences.

We carefully considered the proposed alternative hypothesis that “the male-specific gene sequences detected by the study might reflect common ancestry of the non-recombining sex-determining region of these plants, and not necessarily genes that underwent the male- and female- sterility mutations that were functionally involved in the evolution of separate sexes.” In the end the data reject this soundly – the 14 species share these sex-specific genes with each other, and the genome organization of these genes is quite different from the hermaphroditic ancestor. It seems like a straw man to set this up as an alternative hypothesis only to have it so readily dismissed.

The approach used sounds clever — knowing that these Phoenix palm species have male heterogamety, the study searched for a possible Y-linked region by sequencing males and females of several species to see if they can find sequences that are present only in males. This indeed yielded 1653 16bp kmer sequences present in males of all 13 species where males were studied, and absent in females of all the species, plus another species where no male was obtained. (The sample sizes can be explained more clearly than in the submitted ms; also the statement that “kmers present in females and absent in males were not present in more than 8 species of the genus” is not very clear. I understand that

no such sequence was found in females of all species, but the number of kmers should be mentioned, to compare with the number of male-specific ones — presumably it is a smaller number).

We reworded the text to make it more clear that kmers unique to females were not shared by more than 8 species. We stress that beyond 8 species there are no female-specific kmers, hence no numbers to report. The numbers of female-specific kmers shared among fewer than 8 species can be seen from fig 1a.

The sequences were used to identify BAC clones and thus to identify candidate sex-determining genes. This revealed only a few genes common to males of all species (or maybe just to 6 of them shown in Figure 2) surrounded by highly repetitive sequence. I did not understand Figure 2. The legend mentions several genes (GPAT3, CYP703, LOG and cytidine deaminase or dpB3Y contig) that appear not to be shown in the figure, and the phrase “for comparison see synonymous mutation rates in the same genes” is obscure — are those mutation rates (or perhaps substitution rates are meant) shown somewhere in the ms?

We have modified the figure legend to correct the names of the scaffolds. We also included an explanation of the sequences that are used as controls in the figure. We have changed the text to read “synonymous substitution (dS) rates” to clarify.

It would seem important to analyze these BACs to try and discover all the genes present in them, not just male-specific ones. As presented, it is unclear to me whether several, or even many, other genes are present. Reference number 15 cited by the authors mentions that the “region segregating with gender [presumably in male meiosis] is approximately 26 cM in length” [presumably, in female meiosis] indicating a total physical length possibly as long as 13 Mb or 2% of the plant’s genome”. Certainly a 26 centiMorgan region suggests a physically large region that might include many genes, and I think that readers need to know what proportion of the region has probably been sequenced in this study, and how much remains unexplored, and how many genes might be included in the unexplored regions.

Indeed the non-recombining region in date palm is predicted to be ~13 Mb, however the aim of this manuscript was to identify sequences conserved in all males of the genus rather than just date palm. It is our hypothesis that the 13 Mb of non-recombining sequence in date palm spread out from the original 4 genes identified here during the process of speciation in the genus. The BACs containing sequence conserved in all males of the genus contained the 4 genes mentioned here where the male-specific kmers were fully contained. The sequence scaffold containing the cytidine deaminase did have one additional protein called MAP1 which did not have any male-specific kmers in its surrounding sequence. Sequence surrounding all the genes here was extremely repetitive with 10-20 kb of nested repeats that did not allow the sequences to be extended any further by BAC walking.

We do hope to explore the non-recombining region in the various clades of the species in a future manuscript. For the current manuscript we remain focused on those sequences

present in all males of the genus

The apparently low gene density, based on the BAC sequencing, is consistent with either the region in question having been non-recombining for a long time before dioecy evolved (e.g. a centromeric region), or with enough evolutionary time since Y-linkage evolved that repetitive sequences have accumulated. Under the second possibility, sequence divergence between the X- and Y-linked regions should be large enough that it can be estimated using the coding regions and introns of any genes that are present in both haplotypes, and the divergence ought to be less than between the X-linked sequences and the orthologs in the outgroup species, before dioecy evolved. It would be good to present such results. If there really are no genes other than the small number identified by the search for male-specific sequences, this should be explained, so that readers understand why this analysis was not done (although one gene with copies in both the X- and Y-linked regions was identified, and might be informative). I don't think that there was a complete absence of other genes, as page 9 mentions that "TBLASTN analysis of the male BAC contigs identified sequences such as MAF1 downstream of cytidine deaminase, having sequence similarity to known genes but no male-specific kmers. These genes are likely not critical to dioecy as they have degraded faster among species". And page 10 mentions a *BAG* gene.

I also feel that the four male-specific genes should be more clearly explained. A table would be helpful. The section of the text describing analyses of the male-specific genes that were identified (page 5 says "highly conserved male-specific genes" and doesn't explain why all 4 genes were not examined). Later in the text, a similar restriction appears, but in other places all 4 genes are discussed. To try and understand the results, I summarized them for myself and I attach the crude table.

We have added in the information as Table 2. We do agree that this helps summarize the work better and by doing so makes the results clearer.

The case that best supports the idea that a gene has been lost from the X-linked region (potentially causing females' male-sterility) is the CYP gene the study identified. Similar CYP703 family sequences were found in 11 other plant species, with multiple gene family members presumably found in at least some of them, since 32 proteins were identified.

In Angiosperms, CYP703 appears to be a single member gene family. That is, only one copy has been identified in genomes analyzed to date. Our inclusion of the 32 proteins from 11 species was to ensure that our annotation of the CYP protein should indeed be CYP703 rather than a member of another CYP clan. Our results do show that the gene is indeed a member of the CYP703 family.

There was no sign of complete or partial sequences of *CYP703* in the published date palm female reference genomes or transcriptome data. But this is far from establishing that this gene really is missing from females (admittedly not an easy task), or that its loss led to male-sterility.

We stress that CYP703 is a single member gene family in all angiosperms analyzed to date. Unless the gene was duplicated in date palm it likely means that loss from the female is not compensated for. While it is possible that the female may contain a CYP703 sequence, there are no kmers present that would support this, so presumably it would have diverged significantly in all the species' females while still being retained. Additional evidence against this is that our sequence normalization showed that the male contained a haploid copy of the gene (1N) while other sequences are at 2N (Figure 3d – see N copies by genome).

Assuming that males of all species (and not females) carry the genes identified, the question is how this can be interpreted. If these plants have a physically large region that stopped recombining in their common ancestor, then, given enough evolutionary time, different species' Y-linked regions will have diverged from one another, both in sequence and in their arrangement and gene content, as has occurred in the Y-linked regions of mammalian genomes, and even among different primate species. If this is the situation, then selecting just those sequences that are universally male-specific should identify genes that have been retained because of their essential functions in males, including presumably the male-determining factor. It would be helpful to explain the basic reasoning and the principle of the test employed, which is not explained in the present version (making this clear at an early stage would also remove the need for the first part of the final section).

Complete absence of a gene from female palms would seem surprising, but the text does not outline what might lead to such a situation.

This is why the reviewer's argument (that the male-specific genes simply arose after the two sexes had distinct sex chromosomes with low recombination) would create a straw man argument. Females had to lose function of genes important for formation of male flowers, and the complementarity of functional male flower genes on the Y, and their loss on the X is what drives our story.

Before we can conclude that the study identifies “the changes foundational to dioecy in *Phoenix*”, we should also exclude other possibilities. A Y-linked region that evolved in the common ancestor of a group of related species might have undergone changes that are not related to involvement in male function. The phrase used on p. 10 (“a passenger of the process of sex chromosome evolution”) presumably refers to some such idea, but in my opinion this is too vague, and the thought should be made clear and explicit. A sequence that is absent in females could, for instance, reflect a duplication of a gene, followed by sequence divergence. The sequence might still be present in the females' X-linked region and also potentially elsewhere in the Y-linked region, but divergence could make these copies hard to detect. The results indeed suggest that some genes lack X-linked copies, suggesting that translocations have occurred into the Y-linked region (as the ms itself mentions). Again, such changes might not be part of the initial evolution of separate sexes, as they are thought to appear during the later stages of sex chromosome evolution, e.g. in *Drosophila* species. With a physically large non-recombining region that might be prone to rearrangements, these suggestions are quite plausible.

Cytidine deaminase perhaps is the example we are looking for. It is present on the Y and appears to be functional, but we are not making the argument that it is involved in specifying maleness precisely because there is still a functional copy on the X in females. We have clarified this as an example of a “passenger” and try to make the situation clear.

The evidence against this alternative explanation for the results, and favouring the authors’ interpretation ought (in my opinion) to be described, or at least mentioned, at an early point and not delayed until the Discussion section. This finally appears on p. 9, where the evidence concerning the male-specific genes’ functions in other plants is discussed, which shows that the mutations in the genes identified by this study cause male or female sterility in other plant species. I include this information in the summary table that I mentioned above. If such a table were present in the paper, the authors could draw attention to important ideas by referring to the table, and then the strange organization of the material might be ameliorated, or the table might help the authors arrange their evidence better.

We have included the table (Table 2) and reorganized the manuscript to help the flow of the analysis and inferences.

An important component of the two-gene hypothesis is not mentioned, however. This is that the proposed dominant female sterility mutation that defines the Y haplotype (along with absence of the male-sterility mutation) should increase the fitness of males, while decreasing that of females. If a complete female sterility mutation was involved, this clearly has the latter effect. However, it will not invade a gynodioecious population unless male function is better than that of the hermaphrodites that are present. This may be difficult to test, but the need for a test (and the point that the study does not fully test the 2 gene model) should be mentioned, especially as the ms mentions the sort of advantages that allow females caused my male-sterility mutations to spread. Is a male-enhancing effect of LOG likely?

We have modified the text to emphasize that the Y chromosome was hypothesized to spread into a gynodioecious population because of its advantage over the hermaphrodite. This is presumably through increased fertility of males that harbor the female suppressor gene. We include some evidence from rare bisexual flowers in date palm that show a potential reduction of male fertility in hermaphrodites (Othmani *et al.* 2017).

OTHER COMMENTS

It should be made clear in the Introduction (not the Discussion) that hybridization can occur among different species in the genus, as this conveys some information that the species are quite close relatives. It is also relevant when discussing X-Y divergence between the clades (and within the species, as I suggest should be added).

We have attempted to stress early in the text that species in the genus can hybridize.

Page 6 includes analysis of the synonymous mutation rate (dS) in the four genes for all

Phoenix species compared to the ortholog in oil palm. The oil palm appears not to be explained in the manuscript, so this comparison is difficult to understand. Is this species a non-dioecious (maybe monoecious) outgroup and does it belong to a genus other than *Phoenix* (page 5 mentions *Elaeis*) without explaining this genus?

We have added the term “monoecious” to describe oil palm through the text including in the abstract to better clarify our use of it in comparison to the dioecious *Phoenix*.

This analysis revealed higher synonymous substitution rates in the sex-specific genes, presumably implying acceleration in the dioecious species. The ms says that possible explanations are in the Discussion, but I could not find them.

The Discussion articulates this a bit further now.

The analyses to detect purifying selection are inconclusive. They suggest purifying selection (which would suggest that a gene has retained a function), or perhaps neutrality (which is possible for a gene has undergone a non-functional duplication in the Y-linked region, and is absent from the X). Again, the motivation for the analyses should be outlined.

Thanks for catching this – we have added text to make clear what the intention of the analysis was, and how the end result is still consistent with the rest of the story, but is, as the reviewer suggests, not by itself very conclusive.

Page 6 mentions that the inferred proportions of the time back to common ancestry of the monoecious and dioecious palms for convergence of each gene tree is quite small (the highest value was only 13.4%, for CYP703). It is stated that the low values are consistent with their “arrival” on the Y chromosome, but, at this point, the reader is unaware of the interpretation involving a gene translocating to the Y from other initial genome locations. This seems possible (as mentioned above), but it does not relate closely to the 2 gene hypothesis for the evolution of dioecy that the abstract mentions, and it would be helpful to refer to the Discussion section, where this is discussed.

We have rearranged the text so that the reader can understand the possible movement of LOG to the sex determination region prior to discussion of purifying selection. We have also referred readers to the discussion section where the model will be more thoroughly discussed.

The final paragraph repeats what has been said already, and can be omitted, particularly as some of the text over-states the case. Specifically, it is not correct to say that the “results provide evidence that *Phoenix* passed through a gynodioecious intermediate“ — they can be viewed as consistent with that model.

We have modified the text and toned down the language to use the term “suggest” rather than “provides evidence.”

Several references are not the most appropriate. For example, the originator of the two-gene hypothesis is Westergaard (he reviewed the evidence in its support in 1958. The mechanism of sex determination in dioecious plants. *Advances in Genetics* 9: 217-281).

Another example is that Spigler et al (reference ³⁵) are cited for the statement that (in my shortened wording) “Females could spread in the population through increased female seed production. The 1978 paper by Charlesworth & Charlesworth deals with this, as does Lloyd’s paper, Lloyd DG 1975. The maintenance of gynodioecy and androdioecy in angiosperms. *Genetica* 45: 325-339.

We have added the Westergaard reference both at the beginning and in the discussion. Thank you for catching this oversight. Additionally we have included the Charlesworth 1978 paper where mentioned.

Reviewer #2 (Remarks to the Author):

Sex determination is a major switch in the evolution of dioecious plants and animals. It used to be a major effort and takes many years to identify candidate genes for sex determination. With the rapid advancement of genomic technology, the authors sequenced the male and female genomes of 14 species in the genus *Phoenix* and identified 4 shared male specific genes. Among the four genes, CYP703 and GPAT3 are known to cause male sterility in monocots, and the LOG-like gene is associated with female flower development. Deletions of CYP703 and GPAT3 in X chromosomes are detected across all 14 species, and one deletion would be sufficient to cause male sterility to initiate the transition from hermaphrodite to gynodioecy. The LOG-like gene was translocated into the non-recombining region of the date palm Y chromosome. The authors’ interpretation of a truncated LOG expression silenced the normal function of LOG to suppress carpel development is plausible. The purifying selection of CYP703 and GPAT3 and near neutral divergence of LOG-like gene among *Phoenix* species support their interpretation.

This innovative genomic approach is suitable for moderately ancient sex chromosomes that are ancestral at the genus level, allowing substantial degeneration of the Y chromosome in each species, resulting a small number of shared Y-specific genes across the genus. It might be suitable in young sex chromosomes that were evolved at the species level but shared the same sex determination genes. The indispensable role of CYP703 and GPAT3 in various lipid synthesis pathways leading to a new role of sex determination is a novel finding, emphasizing the importance of gene network for understanding underlining mechanisms of various biological morphologies and functions. Their work is an excellent contribution to the sex chromosome research community, and will be of interest to a broad group of researchers.

A minor change:

The clustering of all X or Y alleles across 14 or 13 species is a major piece of evidence to conclude that sex chromosomes are ancestral in the genus *Phoenix*. For this reason, I

would suggest moving sup figure 5 to the main text.

I indeed believe this is important but as previous published results have shown this (Cherif *et al.* 2015), it simply strengthens the conclusion that the sex chromosomes are ancestral. For the sake of space we will leave it in the supplementary information.

Reviewer #3 (Remarks to the Author):

The manuscript "Genus-wide sequencing supports a two-locus model for sex-determination in Phoenix" includes some interesting data and analyses aimed at understanding the origin and evolution of X and Y sex chromosomes in the date palm genus, Phoenix. The findings, however, are presented in way that is difficult to follow and the conclusions are FAR too speculative. In my opinion the manuscript must be overhauled in order to make the results more accessible and speculations should be reframed as hypotheses to be tested through functional analyses of gene function.

Some general comments and suggestions for improvement include:

1. Do not forget to acknowledge the work of Westergaard (e.g. 1958, Adv Genet) in the introduction and discussion. This work was really the basis for the more formal models developed by Charlesworth & Charlesworth.

Westergaard reference has now been added in the introduction.

2. The results of the comparative K-mer analyses (Figure 1a) are quite impressive and key for identifying the putatively sex-linked BACs and 10X-Genomics scaffolds, but the in my opinion the manuscript belabors the K-mer analysis results. The work could be presented as building on the K-mer analysis results as follows: 1) K-mer analyses implicate Y-specific sequences; 2) Assemblies of reads with Y-specific K-mers used to develop probes for BAC libraries and 10X-Genomics scaffolds; 3) Consensus sequences assembled for Y and X haplotypes for the putatively non-recombining sex determination region of the date palm genome (Figure 3); 4) Annotations of these consensus sequences reveal putative Y-specific genes and structural differences between non-recombining portion(s) of the Y and X chromosomes in date palm; 5) Annotations and phylogenetic analyses of these genes suggest the timing of duplication and loss events relative to the split between oil palm and date palm lineages and origin and early diversification of Phoenix species; 6) Gene homologies provide basis for HYPOTHESIZED gene functions.

We have adjusted the manuscript to follow this order by rearranging sections and text. We have also changed the language to more clearly stress the hypothetical nature of the predicted gene functions.

3. Most of the gene trees are not well resolved. Are they base on the coding sequence

alone? If so, could adjacent non-coding sequence be aligned among Phoenix species and included in the analyses?

Maybe.

4. I did not see characterization of scaffold lengths and the quality of the 10X-genomics assembly. It seems these were important for identifying and characterizing the X and Y haplotypes, so the 10X-Genomics assembly should be described in detail.

We have added scaffold statistics from the 10X Genomics assembly in the Results section.

5. How large in the non-recombining portion(s) of the Phoenix sex chromosomes? Do the assemblies completely span this region? Can recombining segments be identified as the ends of the consensus X and Y assemblies? Are the authors certain they have identified all of the male-specific genes? How about the non-coding regulatory RNAs (e.g. as described for sex determination in persimmon)?

The non-recombining region in date palm is predicted to span ~13 Mb (Mathew et al. 2014). However, for the genus Phoenix it is expected to vary among the clades/species. While we have likely not identified all male-specific genes to date palm or other species, we feel confident we have thoroughly identified genes that are male-specific in all species of the genus. In the future, we do hope to continue the work here to follow the spread of non-recombination (and the addition of male-specific genes) through the process of speciation. However, the goal of this manuscript is to focus on those genes that are only present in all males of the genus while being absent from their counterpart females. While we cannot rule out ncRNAs we do believe that those sequences would be contained within the sequences we identified here due to the requirement they be conserved across all males. We have attempted to modify the text and tone down claims such that alternative pathways for sex-determination such as ncRNAs are considered.

6. How does the data presented in the manuscript allow any inference about the evolution of gynodioecy? Without any data on gene function, how can the authors assert that their data implicate a two-gene sex determination system. How can the authors reject the possibility that one of the inferred Y-specific genes is acting as a master regulator directing gene interactions that suppress female organ development and promote male function.

We have attempted to tone down the language on gene function to say that it is hypothesized and that further functional validation in a Phoenix species will be required to validate the hypothesis. Indeed one of the genes could be a master regulator and the others early additions to the non-recombining region.

In general, the manuscript is poorly organized and many conclusions are not well supported. At the same time, the work does advance understanding of the Y and X chromosome gene content and structure in date palm.

Reviewers' comments:

Reviewer #1 (Remarks to the Author):

This ms is somewhat improved, but still requires work to communicate well. I attach an edited version with many changes that were needed in order to try and understand the authors' meanings in many places.

Major comments

Some logical problems remain.

1. An important example, about which the ms appears slightly confused, is the following. The fact that some feature of the genome (such as a rearrangement), or sequence feature, is found in all Phoenix males investigated is consistent with the possibility that it could have played a role in the evolution of the Y-linked region, but it does not necessarily suggest this. If a completely Y-linked region evolves, rearrangements can occur afterwards, and sequence divergence will also occur. Such observations do suggest complete Y-linkage, but inferring that they were functionally involved in sex-determination is not justified.
2. The section about inference of natural selection from dN and dS values is inadequate. It is unclear what species or alleles were compared (maybe X versus Y, or maybe lineages using outgroups to infer X and Y substitutions). If outgroups were used, it must be explained how diverged they are so that readers can judge whether the divergence is suitable for reliable analyses. Saying "the synonymous substitution rate (dS) in genes" is really inadequate.

The paragraph about assessing whether divergence at synonymous and nonsynonymous sites occurs among the Y-linked genes in the genus Phoenix according to neutral divergence is also inadequate. It is completely unclear what was compared, and only for one gene are both X- and Y-linked copies mentioned. This section should be shortened, but must be made clear if it is included at all (which I think is important, in order to test whether these Y-linked sequences could simply be duplicated sequences into a non-recombining region, a question that ought to be explicitly discussed).

3. The idea that Y-linked genes are deleted from females (early in the Discussion section) is mystifying (as is the information that ", and deletion of the gene in the proto-X would not be compensated for"). A deletion is mentioned in the results ("the CYP703 and GPAT3 deletion"), but is not clearly explained. In writing about male-specific sequences, it is important to make clear whether X- and Y-linked alleles exist (if they are highly diverged, one might detect only one and miss the other, making it look as if one was deleted) or whether good evidence has been obtained that no copy exists on one or other of this chromosome pair. It is unclear whether the

authors have shown that there is ONLY one CYP703 copy in the date palm genome, and it is in the Y-linked region and absent from the female genome. If so, the evidence should be clearly described in the results and a possible explanation should be given — perhaps the idea is that a loss-of-function mutation occurred when females arose, and that this involved a gene deletion — as written, the text is obscure. The section entitled “A model for the evolution of dioecy in the genus Phoenix” merely repeats standard ideas about sex chromosome evolution, and can be greatly shortened and included in the preceding section, as it repeats points that are already made in that section.

4. A chromosomal inversion is mentioned on p. 11, but the evidence for an inversion doesn't seem to be described. Also phrases like “The rearrangement that brought CYP703 to the Y chromosome is probably the same event that also brought GPAT3 and cytidine deaminase” need to be related to clearly described data showing that rearrangements occurred. If rearrangement brought genes to the Y, this conflicts with the idea that their presence on the Y, and not on the X, is due to deletions of copies associated with male-sterility mutations (see above). The ms is currently not consistent. Translocation of the LOG gene into the proto-Y non-recombining region is also speculative, although of course it could have led to the suppression of female flowers in males. However, two important points are left unclear (i) whether the dominance is correct for this to help in the evolution of a Y-linked male-determining region, and (ii) whether such a mutation is likely to have increased male functions. It is very weak simply to state that it “would then spread in the population, possibly through increased fertility of a male over a hermaphrodite by greater pollen production”, without any evidence for such an effect. I suggest a shorter text at this point, that shows the true state of the evidence (if I have understood the text correctly).

MINOR COMMENTS

- 1) Vague or obscure writing should be made more precise and/or clear, for example
 - a) Does “these genes were likely foundational to the evolution of dioecy” mean “these are probably the sex-determining genes”?
 - b) Page 4: how many kmers were unique to females?
 - c) What does 10X Genomics) scaffolds mean?
 - d) How were variants phased?
 - e) “highly similar” is vague — please quantify silent or synonymous site divergence or some relevant measure.
 - f) Page 6: the meaning is obscure of “male-specific LOG kmers in the hermaphroditic palms Brahea and Livistona”
- 2) Sex-determination in the melon is a developmental process, and is not relevant to genetic systems such as the one studied in this ms.

Reviewer #2 (Remarks to the Author):

The authors have addressed my questions, and no further comment.

Reviewer #3 (Remarks to the Author):

The authors have addressed many of my concerns with their initial submission, but there are a few points that I hope they will sharpen before publication. First, the authors' discussion of gene function is unnecessarily speculative in my opinion. Specifically:

L260 and beyond - Have the authors formally tested for differential expression in male and female flowers? For each exon? Given that there is some expression in female flowers, how would differential expression be "consistent with a possible role in suppressing female flower formation in males"?

L161 - What "Analysis" "...indicated that the LOG-like gene is related to the OsLOG5 and OsLOG9"? Supplementary Fig. 3?

L281 - Is LOG "maintained as single copy in males"? This is not consistent with k-mer coverage data nor mRNA data shown in figure 3. How can the female be expressing the putatively male-specific LOG gene? Are the authors certain that the female signal is coming from an autosomal gene? If so, why doesn't the same autosomal gene contaminate the k-mer coverage signal for the male copy? Sorry if I missed an explanation.

L297 - Do the authors mean to say the MAP1 is missing on male of other Phoenix males or could it be autosomal in other species?

"Putative functions of the male-specific genes" section - I appreciate the authors discussing information about the functional data for CYP703, GPAT3 and LOG in other systems, but whereas the section is titled "Putative functions...." the wording seems to imply a greater degree of certainty about function in Phoenix males than is warranted. I suggest that the authors make it very clear that hypothesized functions of CYP703, GPAT3 and LOG in Phoenix remain to be tested.

L339 Onward - The authors have no data to support the ancestor of dioecious Phoenix species was gynodioecious. This is the model of Westergaard and Charlesworth & Charlesworth, but it is presented the authors' model that "agrees well" with the classical two gene model for the origin of dioecy. I suggest the authors remove much of the speculation in the discussion section entitled

"A model for the evolution of dioecy in the genus Phoenix" and simply state that they hypothesize that CYP703, GPAT3, and LOG (but see concerns above) were male specific in the last common ancestor of all Phoenix species and X- and Y-linked Cytidine Deaminase genes have been diverging over the same time period. The discussion of branch-lengths in gene trees (in this section and lines 238-240) and purifying selection (dN/dS is < 1.0 for all of these genes!; failure to reject neutrality does not prove neutrality!) is very weak. Further, the observation that X- and Y-linked *P. rupicola* Cytidine Deaminase genes are sister to each other in the gene tree (Supplementary Fig. 5) is swept under the rug. Is it possible that there has been X-Y recombination has occurred secondarily in *P. rupicola* in the segment including Cytidine Deaminase?

Secondly, the authors' description of figure 3 illustrating structure of the non-recombining the sex determination region needs to be clarified. With some clarification, that figure should summarize the critical findings of this research, but currently I am not sure what is being shown in panel a, and panel d is very confusing. For example: L148-150 - The coverage information is clearly presented in the plot of "N copies (average) track and the "#species" is informative. The "N copies (by genome)" track, however, is quite confusing. What is the Y axis? Why are the gene-associated K-mers more clearly evident in the green track for *Brahea* and *Livistona*? I am guessing that this is because the genes are in 2 copies in these species and one in Phoenix males, but I think this should be clarified for readers.

One last correction for the introduction - Whereas some cucurbits are dioecious, melon is not. The Boualem et al. study experimentally converted monoecious melon to dioecy.

Response to Reviewers' Comments for **NCOMMS-17-24316B**, "Genus-wide sequencing supports a two-locus model for sex-determination in *Phoenix*"

Reviewer #1 (Remarks to the Author):

This ms is somewhat improved, but still requires work to communicate well. I attach an edited version with many changes that were needed in order to try and understand the authors' meanings in many places.

We thank the reviewer for the edited version of the manuscript and have integrated the suggested changes.

Major comments

Some logical problems remain.

1. An important example, about which the ms appears slightly confused, is the following. The fact that some feature of the genome (such as a rearrangement), or sequence feature, is found in all Phoenix males investigated is consistent with the possibility that it could have played a role in the evolution of the Y-linked region, but it does not necessarily suggest this. If a completely Y-linked region evolves, rearrangements can occur afterwards, and sequence divergence will also occur. Such observations do suggest complete Y-linkage, but inferring that they were functionally involved in sex-determination is not justified.

We have attempted to address this concern in two ways. We have changed the language in the manuscript to remove the implication that all features identified as sex-linked are involved in sex determination. We have also clarified that the fact that the genes identified here were likely in the non-dioecious ancestor and yet now are (1) only present in males, (2) conserved across the genus (3) with evidence of selection, and (4) shown to be involved in flower development in other monocots and so are likely involved in their shared phenotype; that being male flowering. While they may be involved in other sex-related functions such as flower density, we believe that the data are most parsimonious with being involved in the shared phenotype. We hope that this combination of changes to the text will satisfy the reviewer's concern.

2. The section about inference of natural selection from dN and dS values is inadequate. It is unclear what species or alleles were compared (maybe X versus Y, or maybe lineages using outgroups to infer X and Y substitutions). If outgroups were used, it must be explained how diverged they are so that readers can judge whether the divergence is suitable for reliable analyses. Saying "the synonymous substitution rate (dS) in genes" is really inadequate.

We have clarified the text here to indicate that in this section, for each analyses, the oil palm was used as outgroup and only Y-linked genes were compared. To reduce any confusion,

analysis of the X-linked Cytidine Deaminase is restricted to the section “Annotation of male-specific sequences” rather than in the section on natural selection.

The paragraph about assessing whether divergence at synonymous and nonsynonymous sites occurs among the Y-linked genes in the genus *Phoenix* according to neutral divergence is also inadequate. It is completely unclear what was compared, and only for one gene are both X- and Y-linked copies mentioned. This section should be shortened, but must be made clear if it is included at all (which I think is important, in order to test whether these Y-linked sequences could simply be duplicated sequences into a non-recombining region, a question that ought to be explicitly discussed).

We thank the reviewer for flagging this section of the manuscript as being in need of clarification, and we have re-written it along the lines suggested. In particular, the specific contrasts are indicated, and the logic of the tests and the inferences are spelled out explicitly. The inference of departure from neutrality arises simply from the expectation that dN and dS would not be different for neutral genomic regions, and these genes do clearly depart from this pattern.

3. The idea that Y-linked genes are deleted from females (early in the Discussion section) is mystifying (as is the information that “, and deletion of the gene in the proto-X would not be compensated for”). A deletion is mentioned in the results (“the CYP703 and GPAT3 deletion”), but is not clearly explained. In writing about male-specific sequences, it is important to make clear whether X- and Y-linked alleles exist (if they are highly diverged, one might detect only one and miss the other, making it look as if one was deleted) or whether good evidence has been obtained that no copy exists on one or other of this chromosome pair. It is unclear whether the authors have shown that there is ONLY one CYP703 copy in the date palm genome, and it is in the Y-linked region and absent from the female genome. If so, the evidence should be clearly described in the results and a possible explanation should be given — perhaps the idea is that a loss-of-function mutation occurred when females arose, and that this involved a gene deletion — as written, the text is obscure. The section entitled “A model for the evolution of dioecy in the genus *Phoenix*” merely repeats standard ideas about sex chromosome evolution, and can be greatly shortened and included in the preceding section, as it repeats points that are already made in that section.

We have modified the text in the results section entitled “Synteny with oil palm”, paragraph 1, to clarify the concept of the CYP703 and GPAT genes having been moved in the male and removed from the female from their ancestral location. Indeed, Figure 3d (“N copies average” panel) and (“N copies by genome” panel) displays that CYP703 is at single copy in all males (i.e. half the average genome coverage) while absent from all females. Furthermore, both date palm females (*Deglet Noor* and *Khalas*) were sequenced to ~70X coverage or 35X per allele. At this depth, there is extremely low probability (effectively 0 from the poisson distribution) that a kmer from CYP703 would be missed. All other females were sequenced to high coverage to ensure the ability to detect those kmers were they present. Altogether this is very strong evidence that the genes are indeed at single copy in males and absent from females.

We have taken the reviewer’s suggestions in the modified manuscript they provided to

further clarify this section regarding deletions of the genes from the females.

4. A chromosomal inversion is mentioned on p. 11, but the evidence for an inversion doesn't seem to be described. Also phrases like "The rearrangement that brought CYP703 to the Y chromosome is probably the same event that also brought GPAT3 and cytidine deaminase" need to be related to clearly described data showing that rearrangements occurred. If rearrangement brought genes to the Y, this conflicts with the idea that their presence on the Y, and not on the X, is due to deletions of copies associated with male-sterility mutations (see above). The ms is currently not consistent. Translocation of the LOG gene into the proto-Y non-recombining region is also speculative, although of course it could have led to the suppression of female flowers in males. However, two important points are left unclear (i) whether the dominance is correct for this to help in the evolution of a Y-linked male-determining region, and (ii) whether such a mutation is likely to have increased male functions. It is very weak simply to state that it "would then spread in the population, possibly through increased fertility of a male over a hermaphrodite by greater pollen production", without any evidence for such an effect. I suggest a shorter text at this point, that shows the true state of the evidence (if I have understood the text correctly).

Regarding discussion of an inversion, we have shortened the section to remove the more speculative statements and included Reviewer 1 edits in the manuscript. We do believe this makes the text clearer and more founded on the current results. Thank you for your guidance in this area. We have adjusted the language to clarify that CYP703 and GPAT3 were moved from their ancestral location based on comparison to oil palm. That is, they do not exist in Date Palm at the location that synteny with Oil Palm would suggest they should. The fact that they still exist only in males, while not at the syntenic location, means that they were moved to their current Y-linked location at some point since the separation with Oil Palm.

MINOR COMMENTS

1) Vague or obscure writing should be made more precise and/or clear, for example
a) Does "these genes were likely foundational to the evolution of dioecy" mean "these are probably the sex-determining genes"?

We have made multiple changes to simplify and clarify the text. At the same time we are trying to not overstate the findings by saying these are the sex-determining genes without functional evidence as per Reviewer 3.

b) Page 4: how many kmers were unique to females?

We have attempted to clarify that there are no kmers unique to females that are shared by more than 8 species. That is, there are no genus-wide female-specific kmers.

c) What does 10X Genomics) scaffolds mean?

We have rearranged the wording to clarify that we are referring to scaffolds sequenced and assembled using 10X Genomics technology.

d) How were variants phased?

The variants are phased by the 10X Genomics assembly software that uses barcoded amplifications of single molecules. We refer readers to the technology for in-depth methods on how the phasing is done.

e) “highly similar” is vague — please quantify silent or synonymous site divergence or some relevant measure

Relevant divergence values are now reported in the text. Additionally, similarity at the DNA level between date palm and oil palm genes has been included (91% identical between cyp in oil palm and date palm, 91% identical between GPAT in oil palm and date palm, 94% identical between LOG (kmer log) and oil palm).

f) Page 6: the meaning is obscure of “male-specific LOG kmers in the hermaphroditic palms Brahea and Livistona”

The text in this sentence has been changed per Reviewer 1 suggestions.

2) Sex-determination in the melon is a developmental process, and is not relevant to genetic systems such as the one studied in this ms.

We have removed melon from the manuscript per Rev 1 and 3 suggestions.

Reviewer #2 (Remarks to the Author):

The authors have addressed my questions, and no further comment.

Reviewer #3 (Remarks to the Author):

The authors have addressed many of my concerns with their initial submission, but there are a few points that I hope they will sharpen before publication. First, the authors' discussion of gene function is unnecessarily speculative in my opinion. Specifically:

L260 and beyond - Have the authors formally tested for differential expression in male and female flowers? For each exon? Given that there is some expression in female flowers, how would differential expression be "consistent with a possible role in suppressing female flower formation in males"?

Indeed, RNA-seq was used in both flower types. Results were normalized by RPKM using two replicates, the expression was significantly different in that genes present only in male flowers were absent from females – that is the statistical results were effectively 0 as there are no reads matching in female flowers or leaves. In the case of male flowers no reads aligned to the Y-specific genes either. Regarding exon level differences, TOPHAT reveals

alignments at the exon however we have not tested for statistical significance of the LOG gene expression levels as we mention in the text that separation of the autosomal and sex-linked copy is not possible when looking only at the cDNA. To separate the autosomal and sex-linked copy requires the surrounding, non-expressed DNA sequence. In the future, deeper sequencing with longer reads may allow separation of the two based on 5'UTR sequence but that is beyond the scope of this paper. We have modified the text to ensure that we the statement that the LOG gene's differential exon expression between male and female is of interest for further research but any statement about its functional role is speculative.

L161 - What "Analysis" "...indicated that the LOG-like gene is related to the OsLOG5 and OsLOG9"? Supplementary Fig. 3?

We have changed "Analysis" to "Our analyses" and clarified text. The analyses we refer to, are the multiple sequence alignment (Clustal W) followed by a phylogenetic analysis inferred by Maximum Likelihood. Details about these analyses are addressed in detail in the methods section under "Phylogenetic analysis".

L281 - Is LOG "maintained as single copy in males"? This is not consistent with k-mer coverage data nor mRNA data shown in figure 3. How can the female be expressing the putatively male-specific LOG gene? Are the authors certain that the female signal is coming from an autosomal gene? If so, why doesn't the same autosomal gene contaminate the k-mer coverage signal for the male copy? Sorry if I missed an explanation.

We observe that the autosomal and male-specific LOG gene have very few differences in their exons, however, the divergence is clear in introns and upstream/downstream regions. Figure 3d ("N copies by genome" and N copies average panels) reveals that these male specific regions (outside of exons) are indeed at 1 copy. The issue is that non-male-specific kmers in the exons would also match to the autosomal copy and so show presence in both males and females. What distinguishes this copy of the LOG gene is that it was identified by using male-specific Kmers to identify longer BAC sequences that span the full Y-linked gene. It is this approach that allows us to separate the autosomal and Y-linked LOG genes, something 16bp kmers cannot do, and show that males contain a single copy of the Y-linked LOG.

L297 - Do the authors mean to say the MAP1 is missing on male of other Phoenix males or could it be autosomal in other species?

We have modified the text in this section. In the BAC contig that contained male-specific Kmers surrounding the Cytidine Deaminase gene we identified, by weak protein similarity, a MAP1 gene that contained no male-specific kmers within it. This would suggest that the MAP1 gene has not been under the same selective forces that the Cytidine Deaminase gene has been. Whether there is an autosomal MAP1 is not clear and beyond the scope of the study as this gene does not contain genus-wide, male-specific kmers that are of highest interest here.

"Putative functions of the male-specific genes" section - I appreciate the authors discussing information about the functional data for CYP703, GPAT3 and LOG in other systems, but whereas the section is titled "Putative functions...." the wording seems to imply a greater degree of certainty about function in Phoenix males than is warranted. I suggest that the authors make it very clear that hypothesized functions of CYP703, GPAT3 and LOG in Phoenix remain to be tested.

We have toned down the language as per Reviewer 1 and 3 to remove the more speculative statements and make clear early in the section that the functions have not been directly tested in Phoenix.

L339 Onward - The authors have no data to support the ancestor of dioecious Phoenix species was gynodioecious. This is the model of Westergaard and Charlesworth & Charlesworth, but it is presented the authors' model that "agrees well" with the classical two gene model for the origin of dioecy. I suggest the authors remove much of the speculation in the discussion section entitled "A model for the evolution of dioecy in the genus Phoenix" and simply state that they hypothesize that CYP703, GPAT3, and LOG (but see concerns above) were male specific in the last common ancestor of all Phoenix species and X-and Y-linked Cytidine Deaminase genes have been diverging over the same time period.

We have significantly shortened and otherwise modified the section "A model for the evolution of dioecy in the genus Phoenix" per Reviewers 1 and 3 comments. Namely we integrated suggestions for the text from Reviewer 1 and cut out major portions from the original text that recapitulated Westergaard and Charlesworth&Charlesworth's model. We do feel that it is important to discuss a possible model for the trajectory to dioecy given the evidence in the results. This is best summarized in Figure 4 and elaborated in this section albeit with the caution that it remains for future testing.

The discussion of branch-lengths in gene trees (in this section and lines 238-240) and purifying selection (dN/dS is < 1.0 for all of these genes!; failure to reject neutrality does not prove neutrality!) is very weak. Further, the observation that X- and Y-linked *P. rupicola* Cytidine Deaminase genes are sister to each other in the gene tree (Supplementary Fig. 5) is swept under the rug. Is it possible that there has been X-Y recombination has occurred secondarily in *P. rupicola* in the segment including Cytidine Deaminase?

The discussion of dN/dS has been re-written, giving more specific information about the contrasts that were done, and the inferences drawn from this analysis are more precisely and compactly articulated. We thank the reviewer for flagging these issues.

Secondly, the authors' description of figure 3 illustrating structure of the non-recombining the sex determination region needs to be clarified. With some clarification, that figure should summarize the critical findings of this research, but currently I am not sure what is being shown in panel a, and panel d is very confusing. For example: L148-150 - The coverage

information is clearly presented in the plot of "N copies (average) track and the "#species" is informative. The "N copies (by genome)" track, however, is quite confusing. What is the Y axis? Why are the gene-associated K-mers more clearly evident in the green track for Brahea and Livistona? I am guessing that this is because the genes are in 2 copies in these species and one in Phoenix males, but I think this should be clarified for readers.

We have reworded portions of Figure 3 caption to better explain the "N copies" panels.

One last correction for the introduction - Whereas some cucurbits are dioecious, melon is not. The Boualem et al. study experimentally converted monoecious melon to dioecy.

We have removed the inclusion of melon in this study as per Reviewer 1 and 3 comments.

Reviewers' Comments:

Reviewer #1 (Remarks to the Author):

As previously, I found this ms interesting. This version is further improved, but still requiring quite a lot of editing, which I have attempted to do on the attached copy. The logic is now reasonably clear, and I have only a few remaining questions, which I entered as comments in the Word file.

Response to reviewers comments for: "Genus-wide sequencing support a two-locus model for sex-determination in *Phoenix*", Torres *et al.*

REVIEWERS' COMMENTS:

Reviewer #1 (Remarks to the Author):

As previously, I found this ms interesting. This version is further improved, but still requiring quite a lot of editing, which I have attempted to do on the attached copy. The logic is now reasonably clear, and I have only a few remaining questions, which I entered as comments in the Word file.

I have integrated Reviewer 1's comments as best possible with the changes tracked in the Word document as "Rev 1". Most of the recommendations have been integrated. I should say that across the review process, Reviewer 1 has made edits on the same text sometimes 2-3 times. I have, therefore, attempted to integrate most of the latest suggested changes while deferring at times to their previous edits that we believe communicate the ideas more clearly. The main recommendations not implemented are 1 – the suggestion to change the term "single gene family" to "single copy gene" when referring to CYP703. The term "single gene family" is more commonly used for the CYP type genes and single-copy would be understood in a different sense. Lastly – the suggestion to remove the last paragraph. We feel that it is important to tie the manuscript into the broader context of Palm sex-determination including the important Oil Palm so only removed a portion.